# The liver and muscle secreted HFE2-protein maintains central nervous system blood vessel integrity

Xue Fan Wang[1,2,13], Robin Vigouroux [1,3,13], Michal Syonov[1,3,13], Yuriy Baglaenko[1], Angeliki M. Nikolakopoulou[4], Dene Ringuette[1,3], Horea Rus [5], Peter V. DiStefano[6], Suzie Dufour[1], Alireza P. Shabanzadeh[1], Seunggi Lee[1,3], Bernhard K. Mueller[7], Jason Charish[1,3], Hidekiyo Harada [1], Jason E. Fish [6], Joan Wither[1], Thomas Wälchli [1,8,9,10], Jean-François Cloutier [11], Berislav V. Zlokovic [4], Peter L. Carlen[1,2,4] & Philippe P. Monnier [1,2,12] ✉

Liver failure causes breakdown of the Blood CNS Barrier (BCB) leading to damages of the Central-Nervous-System (CNS), however the mechanisms whereby the liver influences BCB-integrity remain elusive. One possibility is that the liver secretes an as-yet to be identified molecule(s) that circulate in the serum to directly promote BCB-integrity. To study BCB-integrity, we developed light-sheet imaging for three-dimensional analysis. We show that liver- or muscle-specific knockout of *Hfe2/Rgmc* induces BCB-breakdown, leading to accumulation of toxic-blood-derived fibrinogen in the brain, lower cortical neuron numbers, and behavioral deficits in mice. Soluble HFE2 competes with its homologue RGMa for binding to Neogenin, thereby blocking RGMa-induced downregulation of PDGF-B and Claudin-5 in endothelial cells, triggering BCB-disruption. HFE2 administration in female mice with experimental autoimmune encephalomyelitis, a model for multiple sclerosis, prevented paralysis and immune cell infiltration by inhibiting RGMa-mediated BCB alteration. This study has implications for the pathogenesis and potential treatment of diseases associated with BCB-dysfunction.

Clinical observations have documented that liver diseases are associated with alterations in the Central Nervous System (CNS) that manifest as behavioral changes such as cognitive dysfunction, mood disorders and sleep disturbance[1]. A large body of work suggests that the accumulation of ammonia contributes to neurological pathologies following liver failure[2]. A study by López-Franco et al. showed that patients with history of hepatic encephalopathy show signs of cognitive impairment even after receiving liver transplant[3]. This suggests that ammonia is not solely responsible for the alteration of brain functions observed after liver failure, and that other liver functions/factors may be involved in maintaining brain health.

The CNS is protected by a Blood CNS Barrier (BCB), which is formed by endothelial cells, astrocyte end-feet, and mural cells (pericytes or smooth muscle cells). Alterations of the BCB is a common theme in many CNS conditions including multiple sclerosis (MS), Alzheimer's, and stroke[4], and may expose the brain parenchyma to harmful substances from the blood[5]. The extravasation of toxic blood components such as Fibrinogen, may contribute to neuronal loss observed in MS, Alzheimer's disease and stroke[5]. BCB-disruption has been observed in several liver pathologies, such as liver failure, resulting in brain edema[6]. The mechanisms that link liver function and BCB-integrity are largely unknown. One possibility is that the diseased liver secretes factors such as matrix metalloproteases that are

detrimental to BCB-integrity[6]. Another unexplored possibility is that the healthy liver secretes one or several factors that serve to maintain BCB function.

Repulsive Guidance Molecules (RGMs) are GPI-anchored proteins comprising 3 members: RGMa, RGMb and RGMc/HFE2/Hjv[7]. They exert their biological functions through interaction with the transmembrane receptor Neogenin and shared co-receptors, the Bone Morphogenic Proteins (BMPs)[8]. HFE2 is mainly produced by the liver, where it regulates iron homeostasis, and by the muscles, where it is involved in skeletal muscle differentiation[9]. RGMa and RGMb are expressed in many organs, but studies have mostly focused on their role in the developing and diseased CNS[10]. In the diseased CNS, RGMa blocks axonal regeneration and modulates neuronal death[11–13]. A potential involvement of RGMa in MS was first proposed from a genetic study showing that a single nucleotide polymorphism in RGMa was associated with MS and was positively correlated with changes in the expression of inflammatory cytokines in the cerebrospinal fluid of MS patients[14]. Other studies showed that inhibiting the RGMa-Neogenin pathway abolishes the T cell response in experimental autoimmune encephalitis (EAE), which ameliorates clinical symptoms[15]. Furthermore, RGMa neutralization promotes axon regeneration and functional recovery in animals with EAE[16]. We sought to investigate the role of RGM-proteins on BCB integrity and its role in disease models.

In this work, we sought to investigate the role of RGM-proteins on BB integrity and its role in disease models. We show that RGMc/HFE2 secreted by both the liver and the muscle promotes BBB integrity by neutralizing RGMa. Moreover, we show that a reduction of HFE2 blood levels is involved in EAE pathology and that HFE2 injection restores BCB integrity thereby promoting functional recovery.

## Results

### *Hfe2* liver knock out alters BCB integrity

Using widefield imaging, we observed that *Hfe2* knockout mice demonstrated an intensive leakage pattern, indicative of BCB breakdown (Supplementary Fig. 1a–c). This was unexpected since both our immunohistochemical analysis, the human protein atlas, and the Gene Set Enrichment Analysis data base of the broad institute all indicate that *Hfe2* is not expressed in the brain[7] (Supplementary Fig. 1d). Therefore, this suggested that a non-brain source of HFE2 regulates BCB integrity".

HFE2 is mainly expressed by the liver[17,18]; hence we hypothesized that HFE2 released by the liver into the systemic blood stream regulates BCB integrity. To investigate the role of liver-secreted HFE2 on BCB integrity, we generated *Hfe2*$^{\Delta Alb\text{-}cre}$ (*Hfe2*$^{fl/fl}$; Alb-cre) transgenic mice to genetically ablate HFE2 production, specifically in the liver. ELISA analysis of the serum demonstrated a $56 \pm 8\%$ reduction of HFE2 levels in the serum, confirming that the liver is a major source of serum HFE2 (Supplementary Fig. 1e).

To characterize BCB integrity in detail, we developed a method allowing for 3D representations of BCB-leakages at any given cortical depth (Fig. 1a). Leakages were defined by the extravasation of Lysine-fixable Texas Red-conjugated 70 kDa dextran (TR-dextran). Animals were perfused to remove residual TR-dextran in the vessels. By using CUBIC lipid-clearing, the optically transparent brains were imaged using light-sheet microscopy to map out BCB-leakage throughout the whole brain. To ensure that our perfusion protocol removed all traces of TR-dextran in the intact brain, while also allowing for visualization of BCB-leakages, we performed middle cerebral artery occlusion (MCAO), which is known to open the BCB of one half of the brain while leaving the other half unperturbed. As expected, leakage-staining was seen in the MCAO damaged hemisphere, while the healthy hemisphere remained devoid of any leakage staining (Supplementary Fig. 2a). Having validated our perfusion protocol, we studied the role of liver HFE2 on BCB integrity. In *Hfe2*$^{\Delta Alb\text{-}cre}$ animals, we observed intense

BCB-leakage in superficial as well as deep regions of the brain, whereas *Hfe2*$^{fl/fl}$ control littermates did not display any such leakage (Fig. 1b, Supplementary Fig. 2b-c; Supplementary Movies 1 & 2).

To ensure that the BCB-leakage observed in the *Hfe2*$^{\Delta Alb\text{-}cre}$ animals did not result from a developmental defect, we ablated HFE2 secretion specifically in adult mice. We infected a cohort of adult *Hfe2*$^{fl/fl}$ mice with a liver-directed adeno-associated virus, AAV8-AlbCre[19], or a control adeno-associated virus, AAV8-GFP. Three weeks post-infection, AAV8-AlbCre infected mice displayed significant reduction in HFE2 serum-levels, which was comparable to that of *Hfe2*$^{\Delta Alb\text{-}cre}$ mice (Supplementary Fig. 1e). *Hfe2*$^{fl/fl}$ mice infected with AAV8-AlbCre invariably displayed severe BCB-leakage, which were not observed in the AAV8-GFP infected controls (Fig. 1b). Both *Hfe2* liver-specific knockout models exhibited significant BCB-leakage, as indicated by increased total fluorescence intensity of residual TR-dextran, indicating that HFE2 plays a significant role on maintaining BCB integrity (Fig. 1b). To confirm this phenotype and validate our BCB-assessment method, we performed in-vivo multiphoton imaging of TR-dextran in *Hfe2*$^{\Delta Alb\text{-}cre}$ and AAV8-AlbCre mice along with their respective controls. We confirmed that the observed leakage was occurring within the brain, by imaging the height of the dura relative to the observed blood vessels (Supplementary Movie 3). Following this validation, we did time-lapse imaging over a 40-minute window to record the temporal dynamics of the leakage. As expected, both *Hfe2* liver knockout mouse models displayed obvious extravascular leakage (Fig. 1c, Supplementary Movie 4). Time-series analysis revealed a significant increase in the total TR-dextran extravasation in the knockout animals compared to their respective controls (Fig. 1c) and this extravasation was observed only in larger blood vessels (>24 μm, Supplementary Fig. 1f). The leakage pattern seen in the in-vivo imaging supported what was observed in our light-sheet BCB assessments (Fig. 1b). We subsequently investigated whether BCB-breakdown in HFE2-deficient mice led to accumulation of blood-borne substances in the CNS. We performed staining for blood-borne Fibrinogen, a well-established marker of BCB-breakdown[20]. Immunostaining for fibrinogen and the endothelial marker isolectin revealed abundant fibrinogen deposition around the cerebral vasculature in HFE2-deficient mice, which was significantly elevated compared to controls (Fig. 1d). Finally, we performed intravenous injection of TR-dextran and studied TR-dextran presence in sections (Fig. 1e, Supplementary Fig. 1g)[21]. We observed TR-dextran deposition similar to that observed by others demonstrating BCB disruption in mice[21]. In conclusion, using 4 different experimental techniques we demonstrate that levels of HFE2 are a critical regulator of BCB-integrity.

### Blood vessel alteration does not result from iron accumulation

HFE2 plays a major role in iron homeostasis, and *Hfe2*$^{\Delta Alb\text{-}cre}$ mice display a $52 \pm 13\%$ increase in iron levels in the blood (Supplementary Fig. 4a)[18]. To determine whether iron levels play a role on BCB-leakage in *Hfe2*$^{\Delta Alb\text{-}cre}$ mice, we investigated whether soluble HFE2 could rescue blood vessel dysfunction. We performed HFE2 deletion in the liver by injecting AAV8-TBG-Cre in *Hfe2*$^{fl/fl}$ mice followed by weekly tail vein injection of 20 μg of HFE2. We observed that HFE2 injection restored blood vessel integrity in the CNS (Supplementary Fig. 3), suggesting that iron is not involved in BCB leakage for these animals. In another attempt to evaluate the role of iron in *Hfe2* KOs, we generated *Hfe2*$^{\Delta Acta\text{-}cre}$ (*Hfe2*$^{fl/fl}$; Acta-cre) transgenic mice to genetically ablate HFE2 production specifically in skeletal muscles. Analysis demonstrated that this procedure reduced HFE2 levels by $32 \pm 10\%$ without altering iron levels in the serum (Supplementary Fig. 4a, b)[18]. In *Hfe2*$^{\Delta Acta\text{-}cre}$ animals, we nonetheless observed BCB-leakages using light sheet imaging and in-vivo multiphoton imaging, suggesting BCB-alteration is not the result of increased iron levels (Fig. 1f, Supplementary Fig. 1h, 2b & d; Supplementary Movie 5). Leakages appeared less pronounced than in *Hfe2*$^{\Delta Alb\text{-}cre}$ animals, which may

reflect the lower decrease in blood HFE2 in $Hfe2^{\Delta\Delta Acta\text{-}cre}$ when compared to $Hfe2^{\Delta Alb\text{-}cre}$. To determine whether liver- and muscle- HFE2 were involved in blood vessel integrity in other organs, we performed intravenous injection of Evans Blue and assessed dye extravasation in the brain, heart, muscles, and liver of $Hfe2^{\Delta\Delta Acta\text{-}cre}$, $Hfe2^{\Delta Alb\text{-}cre}$, and $Hfe2^{fl/fl}$ mice. We observed increased Evans blue extravasation in the brain of both $Hfe2^{\Delta\Delta Acta\text{-}cre}$ and $Hfe2^{\Delta Alb\text{-}cre}$ mice, whereas we could not observe any difference between the muscle, heart and liver of KO vs control mice (Supplementary Fig. 5). Together these results indicate that liver- and muscle-secreted HFE2 may play a pivotal role in the maintenance of BCB-integrity.

## Hfe2 muscle-and liver-knock out results in neuronal loss and behavioral deficits

BCB disruption in $Hfe2^{\Delta Alb\text{-}cre}$ animals triggers the extravasation of fibrinogen, which can be toxic to brain neurons[22], hence, we surveyed the number of cortical neurons in $Hfe2^{\Delta Alb\text{-}cre}$, $Hfe2^{\Delta\Delta Acta\text{-}cre}$, and AAV8-AlbCre animals. Interestingly, in both the liver- and muscle-specific $Hfe2$ knockouts, we observed a strong reduction of the number of cortical neurons (Fig. 2a). The observed reduction in the number of neurons in $Hfe2^{\Delta\Delta Acta\text{-}cre}$ animals fits with data showing that a serum pool of HFE2 serves to regulate BCB integrity. Following infection with AAV8-AlbCre, to ablate liver production of HFE2, we also observed a

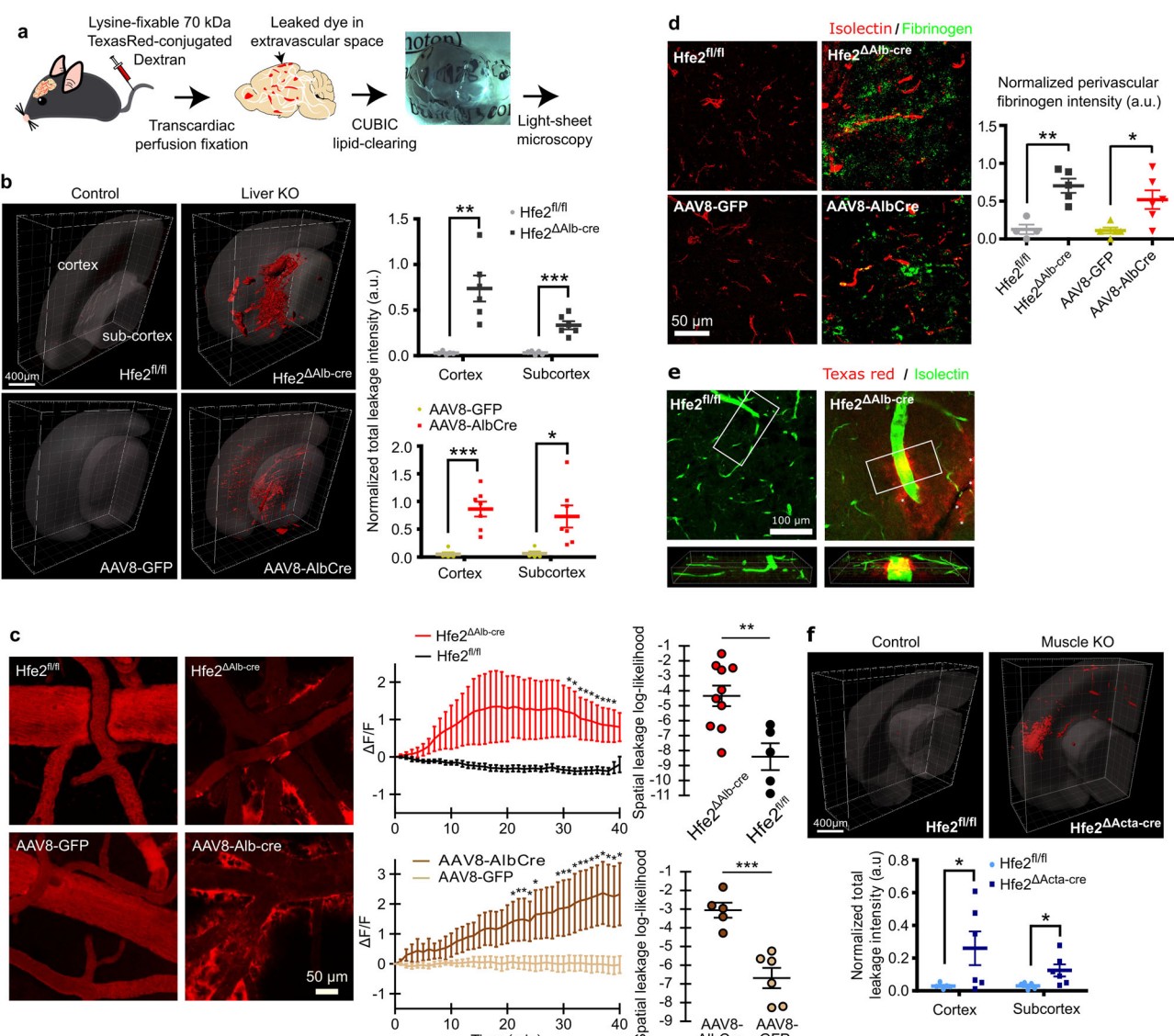

**Fig. 1 | *Hfe2*-deficient mice displays severe BCB disruption. a** Schematic of the BCB assessment method using light-sheet microscopy. **b** 3D rendered representative light-sheet microscopic images of TR-dextran deposition in CUBIC-cleared brains (scale bars, 400 µm) and time-series quantification (mean ± s.e.m.; unpaired two-tail t-test; $Hfe2^{fl/fl}$ (control) $n = 5$, $Hfe2^{\Delta AlbCre}$ $n = 6$, AAV8-GFP $n = 5$, AAV8-AlbCre $n = 7$; n representing data generated from one brain). **c** Representative in-vivo multiphoton images of TR-dextran at 40 mins time-point (scale bar, 50 µm) from the parietal lobes of the cerebral cortex. The normalized extravascular fluorescence (ΔF/F) intensity was plotted over time (mean ± s.e.m.; unpaired two-tail t-tests) and the groups differences were assessed with spatial leakage log-likelihood (repeated measures factorial ANOVA rendered as approximate unpaired two-tail t-test; $Hfe2^{fl/fl}$ $n = 5$, $Hfe2^{\Delta AlbCre}$ $n = 10$, AAV8-GFP $n = 6$, AAV8-AlbCre $n = 5$). **d** Representative confocal images of fibrinogen disposition around the vessels of cerebral cortex (isolectin) of Hfe2-deficient mice (scale bar, 50 µm) and quantification (mean ± s.e.m.; unpaired two-tail t-test; $Hfe2^{fl/fl}$ $n = 4$, $Hfe2^{\Delta AlbCre}$ $n = 5$, AAV8-GFP $n = 5$, AAV8-AlbCre $n = 6$). **e** Confocal imaging of TR-dextran deposition in brain slices from the cerebral cortex with endothelium counterstain. The same observation was made in sections from 3 AAV8-GFP and 3 AAV8-AlbCre brains. **f** 3D rendered representative light-sheet microscopic images of TR-dextran deposition in CUBIC-cleared brains (scale bars, 400 µm; mean ± s.e.m.; unpaired two-tail t-test; $Hfe2^{fl/fl}$ $n = 5$, $Hfe2^{\Delta ActaCre}$ $n = 6$). $*P < 0.05$, $**P < 0.01$, $***P < 0.001$. Source data are provided as a Source Data file (includes exact p-values).

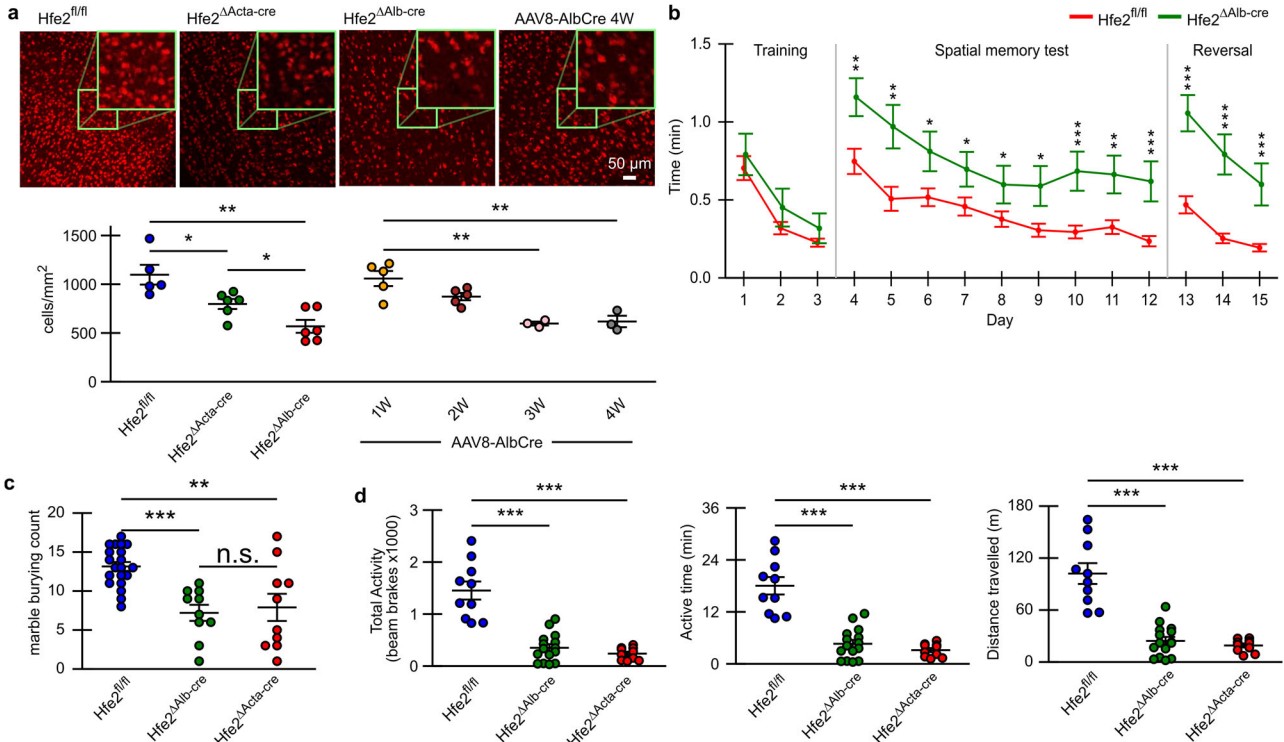

**Fig. 2 | *Hfe2* -liver and -muscle knockout affect neuron number and behavior.**
**a** NeuN staining of the same cortical area (parietal lobes) shows a significant decrease in the number of neurons (scale bar, 50 μm; mean ± s.e.m.; unpaired two-tail t-test). All mice were 6-8 weeks old when experiments were initiated. Replicates *Hfe2*$^{fl/fl}$ n = 6, *Hfe2*$^{Alb-cre}$ n = 6, *Hfe2*$^{Acta-cre}$ n = 6, and AAV8-Alb-cre n = 5, 5, 3, and 3 at weeks 1, 2, 3, and 4 respectively. **b** Hfe2 liver knockout mice display behavioral abnormality. Quantification of escape latency shows greater time response in Liver knockout (mean ± s.e.m.; unpaired two-tail t-test; *Hfe2*$^{fl/fl}$ n = 10 and *Hfe2*$^{ΔAlb-cre}$ n = 5). **c** Bar graph of marbles buried by *Hfe2* KO mice (mean ± s.e.m.; unpaired two-tail t-test; *Hfe2*$^{fl/fl}$ n = 20, *Hfe2*$^{ΔAlb-cre}$ n = 10, and *Hfe2*$^{ΔActa-cre}$ n = 10). **d** Bar graph of three main open field test criteria for *Hfe2* KO mice (mean ± s.e.m.; unpaired two-tail t-test; *Hfe2*$^{fl/fl}$ n = 10, *Hfe2*$^{ΔAlb-cre}$ n = 15, and *Hfe2*$^{ΔActa-cre}$ n = 10). *$P < 0.05$, **$P < 0.01$, ***$P < 0.001$. Source data are provided as a Source Data file (includes exact p-values).

progressive loss of cortical neurons over 3 to 4 weeks (Fig. 2a). Next, we performed a series of behavioral tests to determine whether neuronal loss correlated with functional deficits. We decided to look at the effect of water maze which is normally used to study hippocampal functions because lesions of the parietal lobe or the retrosplenial cortex have also been shown to affect learning in this test[23,24], marble burying and open-field test are normally used to study anxiety and can be used to study the role of cortical lesions on anxiety related behavior[25]. Interestingly, *Hfe2* knockout in the liver resulted in a strong reduction of functional scores in marble burying, open field, and water maze tests, indicating impaired brain functions (Fig. 2b–d). Similarly, *Hfe2*$^{ΔActa-cre}$ animals also displayed functional deficits, these were unlikely to be the result of muscle weakness as the hanging wire strength test did not show any difference between our groups (Supplementary Fig. 6).

**RGMa and HFE2 have opposite effects on Claudin-5, PDGF-B, and HIF-1α expression**
Pericytes are key to maintaining blood vessel-integrity[26,27], hence we studied pericyte coverage of endothelial cells in *Hfe2* liver KO (Fig. 3a). Interestingly, both *Hfe2*$^{ΔAlb-cre}$ and AAV8-AlbCre animals displayed a significant reduction of pericyte coverage when compared to control animals, suggesting that this alteration of pericyte coverage may trigger blood vessel-breakdown (Fig. 3a). PDGF-B expression by endothelial cells is critical for maintaining pericyte coverage. Consequently, we used quantitative PCR on purified endothelial cells to determine whether knocking out *Hfe2* affects mRNA levels for PDGF-B (Fig. 3b). PDGF-B mRNA levels were reduced by ~4.5 ± 1.6× fold in *Hfe2*$^{ΔAlb-cre}$ when compared to *Hfe2*$^{fl/fl}$ mice (Fig. 3b), suggesting that HFE2 contributes to maintenance of blood vessel integrity by

promoting PDGF-B expression. To investigate the mechanisms whereby HFE2 regulates PDGF-B expression, we studied PDGF-B mRNA levels in cultured endothelial cells. Surprisingly, addition of HFE2 to endothelial cultures did not influence PDGF-B mRNA levels (Fig. 3c). We hypothesized that HFE2 may counteract the effect of another factor that negatively regulates PDGF-B expression in endothelial cells (bEnd3 cells). In axonal growth experiments, we have observed that HFE2 blocks the inhibitory activity of RGMa. The addition of RGMa to the laminin-substrate shortens axons when compared to laminin on its own, while the addition of HFE2 to the medium restored axonal length to control levels, suggesting that HFE2 can fully neutralize RGMa (Supplementary Fig. 7). We detected RGMa in human and murine blood serum by Western Blotting (Supplementary Fig. 8) and ELISA ( ~ 0.9 μg/mL), respectively, indicating that endothelial cells are in direct contact with both blood-borne RGMa and HFE2. We addressed the possibility that RGMa and HFE2 – as observed in growing axons – have opposite effects on endothelial cells. RGMa was added to the medium of cultured bEnd3 cells. This induced a 4.1 ± 1.3× fold reduction in PDGF-B mRNA levels, which prompted us to test whether HFE2 positively impacts PDGF-B expression by counteracting this RGMa effect (Fig. 3c). As predicted, when HFE2 was added to RGMa treated cells, we observed an abrogation of the RGMa effect on PDGF-B mRNA levels (Fig. 3c). PDGF-B is present in the matrix of endothelial cells, hence, we evaluated the expression of this protein using immunohistochemistry. In agreement with our mRNA data, RGMa addition to the medium reduced PDGF-B expression, which was rescued by the addition of HFE2 to the medium (Supplementary Fig. 9).

Blood vessel integrity is also regulated by tight junctional molecules, hence, we asked whether the expression of tight junction proteins is altered upon ligand treatment in cultured bEnd3 cells and

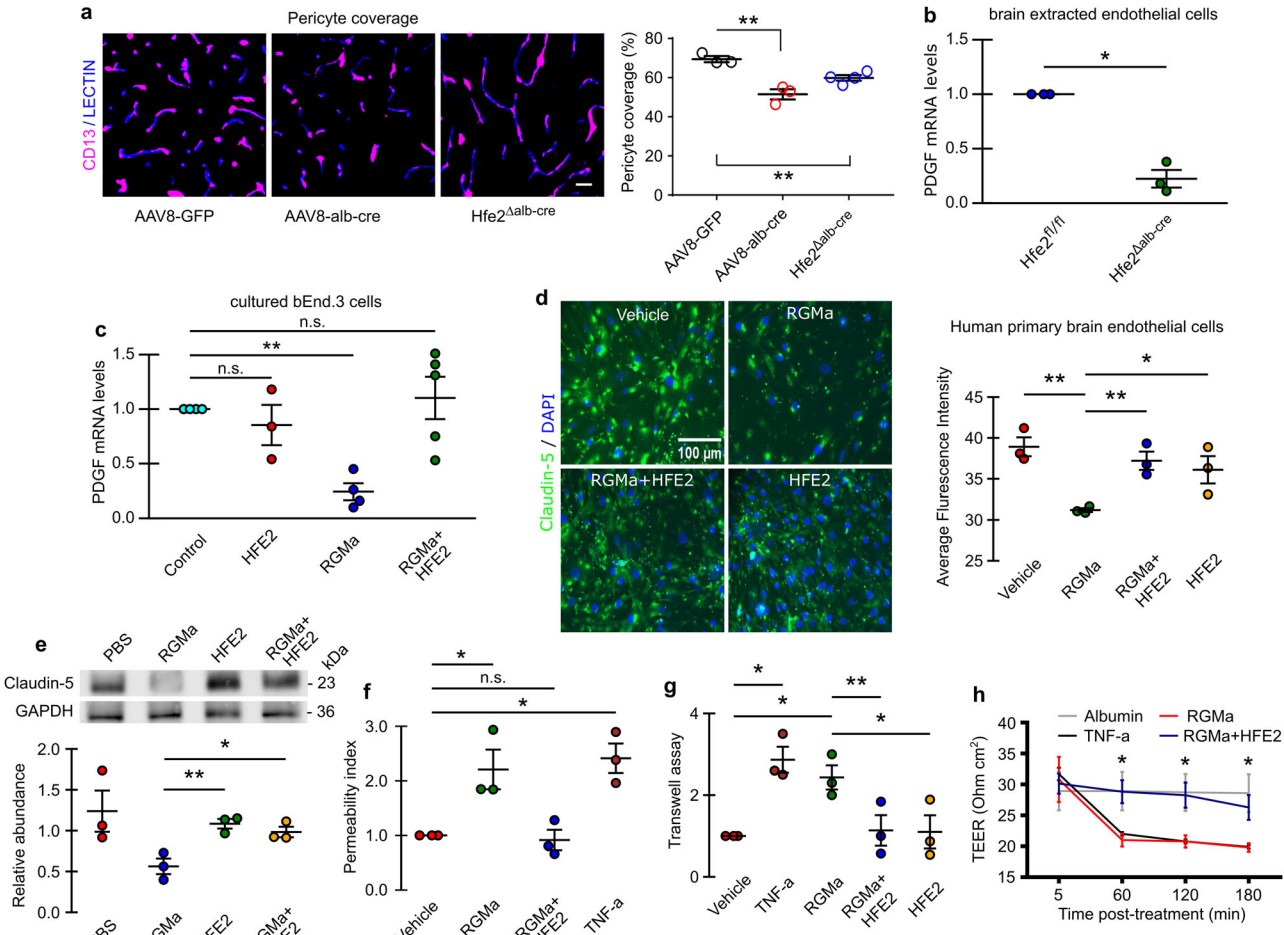

**Fig. 3 | Soluble HFE2 and RGMa effects on endothelial cells. a** Knockout of liver *Hfe2* reduces pericyte coverage in section from the cerebral cortex (mean ± s.e.m.; unpaired two-tail t-test; AAV8-GFP n = 3, *Hfe2ΔAlb-cre* n = 4, and AAV8-AlbCre n = 3). Scale bar, 50μm. **b** Quantification through quantitative RT-PCR shows a reduction in PDGF-B mRNA levels in liver knock out animals (mean ± s.e.m.; paired two-tail t-test; *Hfe2fl/fl* n = 3 and *Hfe2ΔAlb-cre* n = 3). **c** Quantification of PDGF-B mRNA in cultured bEnd.3 cells following indicated treatments. Treatment with RGMa leads to a significant reduction of PDGF-B mRNA levels that is rescued with HFE2 addition (mean ± s.e.m.; paired two-tail t-test; HFE2 n = 3, RGMa n = 4, and RGMa+ HFE2 n = 5; n representing an independent experiment). **d** Immunocytochemistry of claudin-5 in human primary endothelial cell monolayer after PBS, HFE2, RGMa, and RGMa

+HFE2 treatments and quantification (mean ± s.e.m.; unpaired two-tail t-test; replicates n = 3). Scale bar, 100 μm. **e** Western blotting of Claudin-5 expression in bEnd3 cell lysates after indicated treatment (mean ± s.e.m.; unpaired two-tailed t-test; replicates n = 3). **f** Transwell permeability leakage assay performed on a monolayer of bEnd3 cells using HRP (mean ± s.e.m.; paired one-tail t-test; replicates n = 3). **g** Transwell permeability leakage assay performed on a monolayer of human primary endothelial cells using 70 kDa FITC-dextran (mean ± s.e.m.; paired two-tail t-test; replicates n = 3). **h** In vitro TEER analysis shows that RGMa alters the integrity of a monolayer of bEnd3 cells, this is rescued by HFE2 (mean ± s.e.m.; unpaired two-tail t-test; replicates n = 3). *$P < 0.05$, **$P < 0.01$, ***$P < 0.001$. Source data are provided as a Source Data file (includes exact p-values).

human primary endothelial cells. Immunocytochemistry revealed obvious Claudin-5 discontinuity upon RGMa treatment, and such disruption was prevented by HFE2 co-treatment (Fig. 3d, Supplementary Fig. 10). Western blot further confirmed a significant reduction in Claudin-5 expression in RGMa-treated bEnd3 cells compared to HFE2 co-treatment and controls (Fig. 3e). RGMa did not appear to impact the expression of Occludin, (Supplementary Fig. 11a). Furthermore, we assessed the expression of a few factors that are known to trigger blood vessel dysfunction. Interestingly, we observed that HIF-1 α, a factor that is regulated by hypoxia and triggers BCB opening[28,29], is upregulated in the presence of RGMa, and that addition of HFE2 to the medium blocks this upregulation (Supplementary Fig. 12). Other factors involved in blood vessel integrity such as PLVAP[30] AKT[30,31] (Supplementary Fig. 11b, c), and YAP[32] (Supplementary Fig. 13), were not influenced by the addition of RGMa to the medium. Hence our results indicated that RGMa treatment on cerebral endothelial cells significantly alters the expression of Claudin-5, HIF-1 α and PDGF-B, which can be completely prevented by HFE2.

## RGMa and HFE2 have opposite effects on endothelial cells and BCB integrity

To further assess the effect of RGMa and HFE2 on BCB integrity, we performed a Transwell permeability assay to study the extravasation of horse radish peroxidase (HRP) and 70 kDa FITC-dextran through a monolayer of bEnd3 cells upon ligand treatment. We found that soluble RGMa significantly increases the monolayer permeability to HRP and FITC-dextran to an extent comparable with TNF-α, which was used as a positive control. The effect of RGMa on the bEnd3 monolayer was significantly attenuated by the addition of HFE2 (Fig. 3f, Supplementary Fig. 10c). Primary human endothelial cell were found to respond similar to bEnd3 cells (Fig. 3g). Additionally, in-vitro assessment using trans-endothelial electrical resistance (TEER) confirmed that RGMa significantly reduced bEnd3 barrier function which was suppressed upon HFE2 co-treatment (Fig. 3h).

The data presented above suggests that RGMa alters endothelial barrier function in vitro. To address this in vivo, we intravenously administered either soluble RGMa alone or with HFE2 into wild type

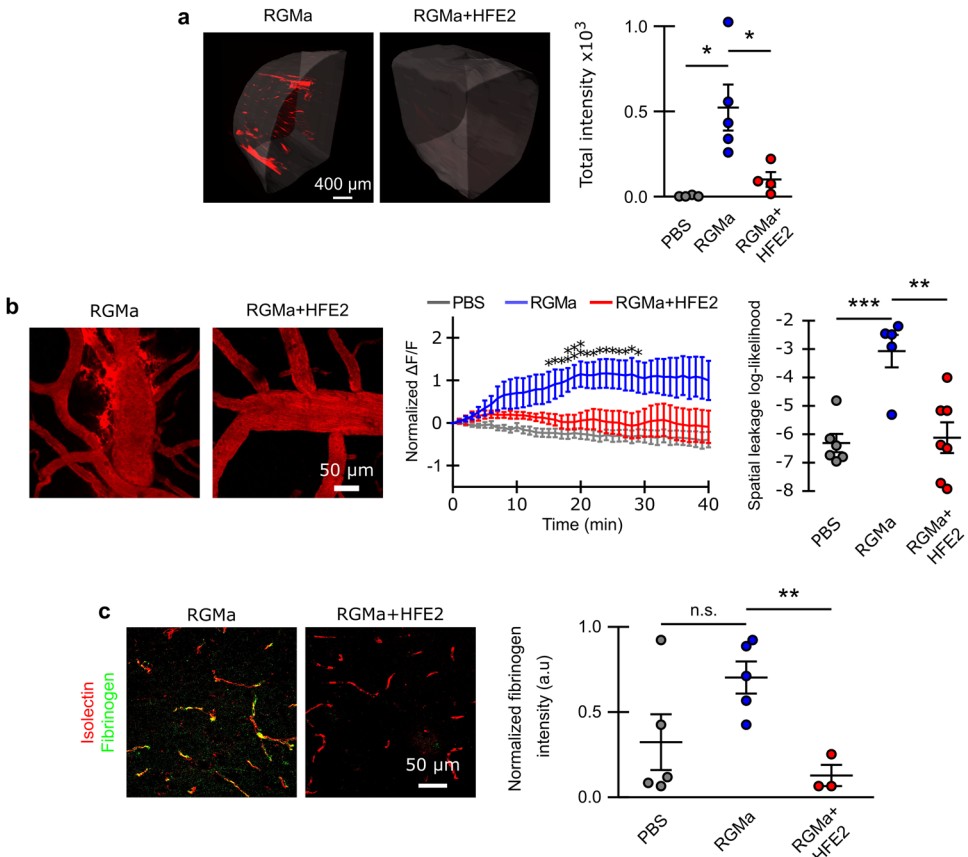

**Fig. 4 | HFE2 and RGMa have opposite effects on BCB integrity. a** 3D rendered representative light-sheet microscopic images of RGMa and RGMa+ HFE2 treated brains (scale bar, 400 μm) and quantification (mean ± s.e.m.; unpaired two-tail t-test; PBS *n* = 3, RGMa *n* = 5, RGMa+Hfe2 *n* = 4). **b** Representative in-vivo multi-photon images from the cerebral cortex of RGMa and RGMa+ HFE2 treated animals. The normalized extravascular fluorescence (ΔF/F) intensity was plotted over time (mean ± s.e.m.; unpaired two-tail t-tests) and the group differences were assessed with spatial leakage log-likelihood (repeated measures factorial ANOVA rendered as approximate unpaired two-tail t-test; PBS *n* = 6, RGMa *n* = 5, RGMa+ HFE2 *n* = 7). **c** Representative confocal images of fibrinogen disposition around the vessels of the cerebral cortex (isolectin) of RGMa and RGMa+ HFE2 treated mice and quantification (mean ± s.e.m.; unpaired two-tail t-test; PBS *n* = 5, RGMa *n* = 5, RGMa+ HFE2 *n* = 3). *$P < 0.05$, **$P < 0.01$, ***$P < 0.001$. Source data are provided as a Source Data file (includes exact p-values).

(WT) mice and assessed BCB-integrity 24 h post-injection. Using the light-sheet BCB assessment tool, we observed intense accumulation of TR-dextran in the brains of RGMa-treated WT mice, indicating that RGMa-injection severely disrupts BCB function (Fig. 4a). Co-treatment of RGMa and HFE2 significantly prevented RGMa-mediated BCB-breakdown (Fig. 4a, Supplementary Fig. 7). Multiphoton imaging further confirmed the disruptive role of RGMa, which displayed a significant increase in TR-dextran extravasation. BCB-integrity in WT animals co-treated with both RGMa and HFE2 were comparable to the PBS treatment (Fig. 4b, Supplementary Movie 6). RGMa-treated WT animals exhibited a significant amount of perivascular fibrinogen accumulation, which was completely abrogated by HFE2 co-treatment (Fig. 4c). Furthermore, we looked at BCB-leakage using widefield imaging, confirming that RGMa induces leakage that is not observed when HFE2 is co-injected (Supplementary Fig. 14). Together, these data identify RGMa and HFE2 as regulators of PDGF-B and BCB integrity.

**Neogenin is involved in RGMa mediated opening of the BCB**
HFE2 is a Bone Morphogenic Protein (BMP) co-receptor, and RGMa has been shown to regulate **BCB** integrity via a BMP-receptor/YAP pathway[33]. To investigate the mechanisms whereby HFE2 regulates blood vessel integrity, we tested whether HFE2 had any effect on the canonical BMP down-stream effector Smad1/5/8, however, we could not observe any significant difference between HFE2 treated cells and controls (Supplementary Fig. 15). Moreover, we assessed YAP activation in a reporter assay and in Western Blots. In these experiments, the

addition of Hfe2 to the medium of bEnd3 cells did not lead to any change in YAP expression and activation (Supplementary Fig. 13).

RGMa interacts with Neogenin to inhibit axonal growth[34]. Although Hfe2 also interacts with Neogenin, HFE2 does not inhibit outgrowth, but rather suppresses the RGMa inhibition of axonal growth (Supplementary Fig. 7). This outgrowth data suggests that HFE2 prevents the interaction between RGMa and Neogenin. To address this possible role for HFE2, we developed an assay in which soluble Neogenin (AP-tagged) will interact with RGMa. We observed that soluble Neogenin-AP interacts with an ELISA-plate coated with RGMa (Fig. 5a). In a competitive binding assay, we show that HFE2 and RGMa significantly prevented the binding of Neogenin-AP to RGMa (Fig. 5a). Also, we show that Neogenin-AP binds to HFE2 and that this interaction is blocked by both RGMa and HFE2 (Fig. 5a). Next, we studied Neogenin expression in blood vessels, and show that it is strongly expressed in endothelial cells (Fig. 5b). Neogenin staining was observed in the lumen of human and murine blood vessels suggesting that it interacts with blood proteins (Fig. 5b). The presence of Neogenin in endothelial cells raised the possibility that HFE2 restores BCB integrity by preventing RGMa binding to Neogenin. A hypomorphic allele with ~90% loss of Neogenin has been described previously, and shows that mice die around P28[35]. Hence, to study the involvement of Neogenin in RGMa-induced BCB breakdown, we used $Neo^{\Delta Tie2-creERT2}$ ($Neo^{fl/fl}$; Tie2-creERT2) knockout animals in vivo, wherein Neogenin is genetically deleted from endothelial cells upon Tamoxifen induction. Immunoblotting on isolated cerebral endothelial cells of $Neo^{\Delta Tie2-creERT2}$

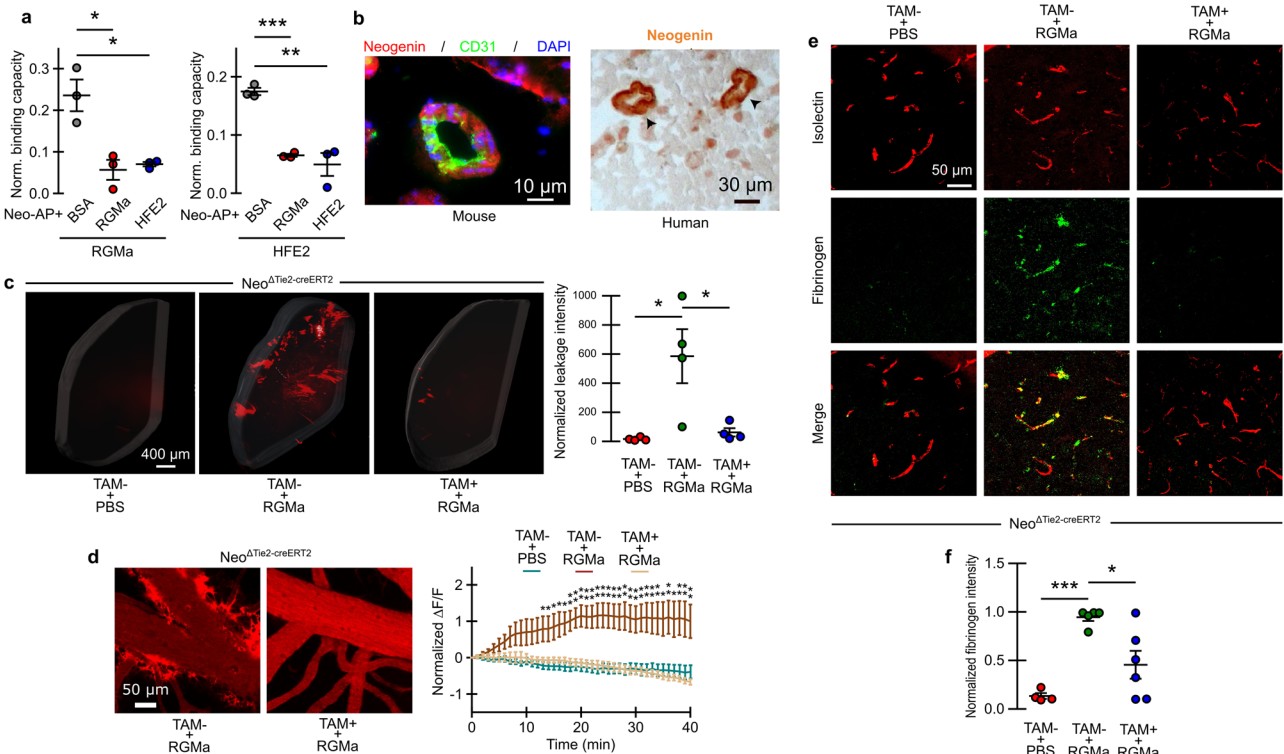

**Fig. 5 | RGMa disrupts BCB through Neogenin receptor. a** HFE2 blocks RGMa-interaction with Neogenin. Competitive binding between RGMa and HFE2 on Neogenin-AP. RGMa blocks binding to HFE2 and HFE2 to RGMa (mean ± s.e.m.; unpaired two-tail t-test; replicates $n = 3$). **b** Expression of Neogenin in the cerebral endothelial cells (CD31) in mice (scale bar, 10 μm) and vascular structure in human (arrows; scale bar, 30 μm). This pattern was observed in sections from 3 mouse brains and 2 human patients. **c** Representative 3D light-sheet images of $Neo^{ΔTie2-creERT}$ mice treated (TAM + ) and untreated (TAM-) with Tamoxifen before subsequent RGMa treatment (scale bars, 400 μm) and quantification (mean ± s.e.m.; unpaired two-tail t-test; TAM- & PBS $n = 4$, TAM- & RGMa $n = 4$, TAM+ & RGMa $n = 4$). **d** Representative in-vivo multiphoton images at 45 mins time-point (scale bar, 50 μm) and time-series quantification (mean ± s.e.m.; two-way ANOVA with post-hoc Bonferroni test; TAM- & PBS $n = 4$, TAM- & RGMa $n = 4$, TAM+ & RGMa $n = 4$). Representative confocal images of fibrinogen staining in the brain (**e**; scale bar, 50 μm) and quantification (**f**; mean ± s.e.m.; unpaired two-tail t-test; TAM- & PBS $n = 4$, TAM- & RGMa $n = 5$, TAM+ & RGMa $n = 6$). *$P < 0.05$, **$P < 0.01$, ***$P < 0.001$. Source data are provided as a Source Data file (includes exact p-values).

revealed complete deletion of Neogenin (Supplementary Fig. 16). Soluble RGMa treatment applied to tamoxifen-treated $Neo^{ΔTie2-creERT2}$ mice did not induce BCB-leakage as observed with WT mice, indicating that RGMa-mediated BCB breakdown is primarily mediated by Neogenin (Fig. 5c, Supplementary Fig. 16). Multiphoton imaging as well as fibrinogen staining in RGMa-treated $Neo^{ΔTie2-creERT2}$ mice also indicated that Neogenin ablation in cerebral endothelial cells significantly prevents RGMa-mediated BCB alterations (Fig. 5d–f, Supplementary Movie 7). These results demonstrate the role of the Neogenin receptor in mediating the effect of RGMa on BCB integrity.

If soluble HFE2 prevents blood vessel alteration by preventing the interaction between RGMa and Neogenin, the neutralization of Neogenin should restore blood vessel integrity in $Hfe2^{ΔAlb-cre}$ mice. Hence, we investigated the role of Neogenin by performing weekly intraperitoneal injections of the Neogenin neutralizing peptide 4Ig[36] in AAV8-TBG-Cre treated $Hfe2^{fl/fl}$ mice (Supplementary Fig. 3). Importantly, we observed that 4Ig treatment significantly restored BCB integrity suggesting that the HFE2 effect on endothelial cells is mediated by Neogenin.

**The RGM/Neogenin pathway modulates BCB integrity in model mimicking aspects of MS**

RGMa is upregulated in MS, a disease manifesting severe blood vessel disruption in the CNS[15,16]. Furthermore, inflammation has been shown to reduce HFE2/RGMc expression[37]. Therefore, we explored the possibility that the BCB-breakdown caused by RGMa/HFE2 imbalance may be a critical component in MS pathology. To test this hypothesis, we used EAE, an animal model mimicking

aspects of MS[15,38] Animals received a MOG peptide that induces paralysis within 2-3 weeks. ELISA and Western Blotting showed that HFE2-levels are strongly downregulated in the serum of EAE mice when compared to sham induction mice (Fig. 6a). Interestingly, RGMa levels were increased in EAE animals (Fig. 6a) and we observed high levels of RGMa around blood vessels within the plaques found in the brain of MS patients (Supplementary Fig. 17). We reasoned that low HFE2 combined with high RGMa levels open the BCB in EAE and that administrating HFE2 should restore the RGMa/HFE2 balance and prevent RGMa-mediated BCB opening. Treatment with soluble HFE2 to EAE mice significantly delayed disease onset and significantly reduced disease severity (Fig. 6b). This effect was suppressed by co-injecting RGMa with HFE2, which fits with the opposite activities of RGMa and HFE2 shown earlier (Figs. 3 and 4). Because $Hfe2$-liver knock out reduces HFE2 levels in the serum and alters BCB-integrity, $Hfe2^{ΔAlb-cre}$ mice should present an aggravation of EAE symptoms. Indeed, $Hfe2^{ΔAlb-cre}$ mice displayed earlier disease onset and significantly increased disease severity at early stages of the disease when compared to controls (Fig. 6c). These data suggest that circulating HFE2 prevents functional impairment by blocking the detrimental effects of RGMa following EAE induction. Because Neogenin mediates the RGMa effect on BCB (Fig. 5), we tested whether Neogenin knock-out in endothelial cells prevents EAE symptoms. As expected, $Neo^{ΔTie2-creERT2}$ mice exhibited significant improvement following EAE induction (Supplementary Fig. 18), which fits with a model in which Neogenin mediates the RGMa/HFE2 effect. Because Tie2 and Neogenin are both expressed by immune cells, we tested whether treatment of immune cells with HFE2 had

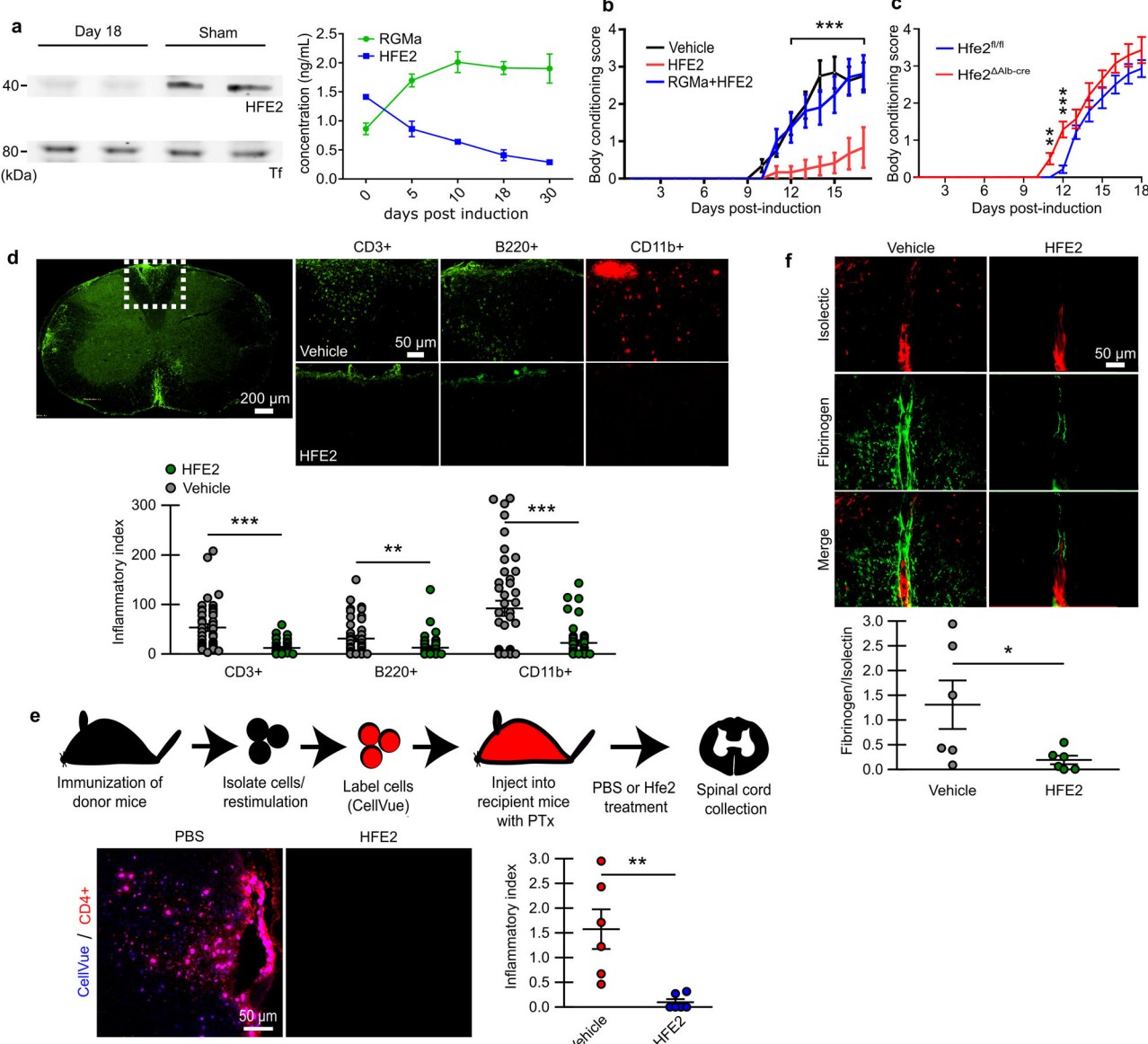

**Fig. 6 | HFE2 administration significantly improves EAE disease outcome.**
**a** Western Blot analysis shows a strong HFE2 reduction in EAE animal serum when compared to control. ELISA quantification showing HFE2 decrease and RGMa increase in EAE animal serum (mean ± s.d.; EAE $n = 3$ and RGMa $n = 3$). **b** Body conditioning assessment ('0' no paralysis to '4' forelimb weakness) of EAE-induced mice with HFE2 +/- RGMa treatment every 3 days (d3, 6, 9, 12 and 15; mean ± s.e.m.; unpaired two-tail t-tests; PBS $n = 8$, HFE2 $n = 6$, RGMa+ HFE2 $n = 8$). **c** Body conditioning assessment of EAE-induced $Hfe2^{fl/fl}$ and $Hfe2^{\Delta Alb-cre}$ mice (mean ± s.e.m.; unpaired two-tail t-tests; $Hfe2^{fl/fl}$ $n = 8$ and $Hfe2^{\Delta Alb-cre}$ $n = 8$). **d** Representative con-focal images of CD3 + , B220+ and CD11b+ immune staining in spinal cord of EAE-

induced mice 3 weeks after induction and quantification (mean ± s.e.m.; unpaired two-tail t-test; PBS $n = 7$, HFE2 $n = 6$ mice; technical replicates shown). **e** Schematic of activated immune cell adoptive transfer. Representative confocal images of CD4-positive labelled transferred cell (CellVue) in PBS-treated or HFE2-treated recipient WT mice and quantification (mean ± s.e.m.; unpaired two-tail t-test; PBS $n = 6$, HFE2 $n = 6$). Ptx; Pertussis Toxin. **f** Representative confocal images of fibrinogen in the spinal cord of HFE2-treated EAE-mice and quantification (mean ± s.e.m.; unpaired two-tail t-test; PBS-treated $n = 6$ and HFE2 $n = 6$). *$P < 0.05$, **$P < 0.01$, ***$P < 0.001$. Source data are provided as a Source Data file (includes exact p-values).

any effect on the expression of immune marker. Treatment with HFE2 had no effect on (i) the adhesion properties of naïve T and B cells, (ii) naïve antigen presenting cells, (iii) naïve immune cell populations, (iv) on activated immune cells, and (v) antigen specific immune cells which also indicates that the HFE2 effect is independent of immune cell priming (Supplementary Fig. 19a–e). Together these data suggest that prevention of EAE symptoms in $Neo^{\Delta Tie2\text{-}creERT2}$ mice results from a restoration of the BCB and not from an alteration of immune cell activation.

Examination of cell infiltrates within the spinal cord of the HFE2-treated EAE mice showed significant reduction of cellular infiltrates as assessed using the H&E staining inflammatory index, where 0 is no

infiltrates and 4 is cellular infiltration within the grey and white matter (Supplementary Fig. 19). Immune cells play a critical role in CNS damage in EAE[39], hence we examined immune cells infiltrations in spinal cord. HFE2-treated mice showed significant reduction in the pan T cell marker CD3 + , the pan B cell marker B220 + , as well as the myeloid marker CD11B+ when compared to vehicle-treated controls (Fig. 6d). To ensure that reduced immune cell infiltration did not result from an effect of HFE2 on immune cell activation, we performed adoptive transfer of activated immune cells into the recipient mice (Fig. 6e). HFE2 treatment once again prevented the infiltration of immune cells into the CNS indicating that the positive effect of HFE2 on EAE progression was likely independent of any effects on immune

cell priming (Fig. 6e). Taken together with the data showing that HFE2 has no effect on the priming of immune cells (Supplementary Fig. 19), this suggests that HFE2 prevents immune cell infiltration by preventing BCB-alteration following EAE induction. A role of HFE2 on BCB integrity was further confirmed by a significant reduction in endogenous fibrinogen extravasation into the CNS of EAE mice treated with HFE2 (Fig. 6f, Supplementary Fig. 19f). Taken together, our data validate a role for HFE2, RGMa, and Neogenin in BCB-maintenance and highlight their importance in MS pathology.

## Discussion

This study defines a role of liver- and muscle-secreted HFE2 that acts on endothelial cells of the Blood-CNS-Barrier to prevent RGMa/Neogenin mediated opening of the BCB. We found that knocking out *Hfe2* in either the liver or the muscle opens the BCB leading to the extravasation of fibrinogen, a reduction of the number of cortical neurons as well as behavioral deficits. We demonstrate a molecular interplay where HFE2 and RGMa compete for binding to Neogenin to modulate the expression of Claudin-5 and PDGF-B in endothelial cells. Finally, we show that RGMa and HFE2/RGMc in the serum have diametrically opposed functions in EAE pathology: RGMa opens the BCB, thereby contributing to a worsening of EAE-symptoms, whereas HFE2/RGMc closes the BCB and prevents paralysis.

### Liver/muscle HFE2 maintains BCB-integrity

Liver-secreted molecules have been hypothesized to play a disruptive role in BCB in multiple liver pathologies[40]. For instance, blood levels for matrix metalloproteinase 9 (MMP9) are increased following hepatic failure and MMP9 blocking with a monoclonal antibody can protect the BCB[6]. Similarly, it has been suggested that cytokines produced by the liver trigger blood brain barrier opening. The liver secretes TGFβ following acute liver failure that directly alters BCB integrity[40]. To our knowledge a demonstration that the liver is directly secreting TGFβ and MMP9 to trigger BCB opening is still missing, as the study of the effect of liver KO of these factors remains to be done.

Moreover, models for a role of the liver on BCB opening suggests that factors secreted by the diseased liver negatively regulate BCB integrity. Here, we demonstrate that the healthy liver can also contribute to BCB maintenance. Indeed, suppression of HFE2 secretion by the liver in healthy animals abrogates BCB integrity. Our surprising findings that a liver-secreted protein plays a key role in the maintenance of the highly specialized and tightly regulated CNS vasculature[41–43] warrants further investigation. For instance, it remains to be shown whether in addition to HFE2, any other liver protein is involved in blood vessel integrity.

We also show that HFE2 secreted by muscles also play a role in BCB homeostasis. A recent article reveals that HFE2 is downregulated in models for Duchenne muscular dystrophy (mdx mouse)[33]. Duchenne muscular dystrophy has been linked to blood barrier breakdown in human patients and mdx mice[33]. However, dystrophin, the gene involved in Duchenne disease is also expressed in the CNS, where its down-regulation may cause BCB alterations[33,44] and it is not possible to directly link BCB alterations in Duchenne to muscular dystrophy. Exercise has many beneficial effects on brain health and helps to restore and maintain cognitive function. It was recently shown that physical exercise regulates cerebral angiogenesis via the release of lactate, which binds to the lactate receptor HCAR1[45]. The HCAR1 receptor is highly enriched in pial fibroblast-like cells, which line the vessels supplying blood to the brain, and in pericyte-like cells, along intracerebral microvessels[45]. The fact that exercise is sensed by the brain (vasculature) suggests that muscle-induced peripheral factors enable direct crosstalk between muscle and brain function. Our data showing that muscle- HFE2 promotes BCB protection, may well identify a critical factor involved in such crosstalk.

### The RGM/Neogenin pathway is involved in experimental models for MS

The cause for MS remains elusive; however, it is believed to have a multifactorial origin comprising a combination of genetic, environmental and stochastic factors[46]. A previous report suggested a genetic association between MS and RGMa. Furthermore, RGMa expression is increased in the CSF of MS patients and lower RGMa levels in the CSF correlate with better functional outcome[47]. In addition, the RGMa-selective antibody ABT-555 promotes functional recovery in animals that received focal EAE[16]. To add further evidence for a role of RGMa in MS, a recent study demonstrates that RGMa is expressed by endothelial cells and triggers BBB dysfunction via a mechanism involving YAP and BMPs[32]. This seems to contradict our data showing that neither YAP nor the canonical BMP pathway are involved in RGMa mediated dysfunction of blood vessels. However, the study from Zhang et al., shows that RGMa is expressed by endothelial cells and that silencing RGMa expression in endothelial cells restores BBB integrity while our study shows that blood borne RGMa is acting on endothelial cells to alter blood vessel integrity. In the study from Zhang et al., RGMa acts on endothelial cells in a cell autonomous manner[32], while our study involves non-cell autonomous mechanisms. This is not the first evidence that RGMa may regulate the same biological function via cell- and non-cell autonomous mechanisms[36]. Indeed, we have shown that RGMa is expressed by growing axons and by cells surrounding these axons, and that both source of RGMa regulate axonal growth[36]. We have identified distinct RGMa domains that regulate cell autonomous and non-cell autonomous axonal inhibition. Moreover, we showed that RGMa inhibits axonal growth via different pathways. Hence, like observed in growing axons, our data indicates that different intracellular pathways mediate cell- and non-cell autonomous alteration of blood vessel integrity by RGMa.

Thus far, HFE2 has not been associated with MS. Our data suggests that liver- and muscle-secreted HFE2 is an important contributor to EAE pathology and advocate for a previously unappreciated role for Hfe2 and the liver (and possibly muscle) in MS progression.

The molecular interplay between HFE2 and RGMa underlying BCB maintenance not only provides a therapeutic approach in numerous diseases manifesting severe BCB disruption, such as Alzheimer's disease[48,49], stroke[50], Parkinson's disease[49], and brain tumors[51,52], but also can be exploited as a switch for transient opening of BCB allowing the efficient delivery of treatment drugs into the CNS.

## Methods

Our research complies with all relevant ethical regulations; naming the board/committee and institution that approved the study protocol.

### Animals

All animal handling was approved according to the Animal Resource & Care Committee at University Health Network (AUP# 6053). Liver-specific Hfe2 congenital knockout, homozygous Hfe2$^{\Delta Alb-Cre}$ transgenic mice (Mutant Mouse Resource & Research Center, University of California, Davis, MMUCD strain 036464-UCD) were maintained on C67Bl/129S1 mixed background and 6-8 weeks mice were selected for experiment. Hfe2$^{fl/fl}$ littermates were used as controls. Deletion of liver Hfe2 production in adult mice was achieved by injecting 6-8 weeks of Hfe2$^{fl/fl}$ were with an average dose of $5 \times 10^{11}$ genome copies of either AAV8-AlbCre (VB1570), AAV8-TBGCre (VB1724) or AAV8-GFP (VB1743) viruses (Vector Biosystems Inc) via tail vein. FC/IO #:410010959. Neo$^{fl/fl}$, was kindly provided by Dr. Jean-Francois Cloutier. They were maintained on C57BL/6 J background. Endothelial specific inducible knock-out mice, Neo$^{\Delta Tie2-creERT2}$, were generated by crossing Neo$^{fl/fl}$ with Tie2 Cre-ERT2 mice (Jackson Laboratories). Conditional knock-out mice, Neo$^{\Delta Tie2-creERT2}$ TAM + , were given 75 mg/kg of tamoxifen between 6-8 weeks of age over 5 consecutive days via IP injection (20 mg/mL corn oil solution). Control mice, Neo$^{\Delta Tie2-creERT2}$ TAM-, were given an

equivalent amount of corn oil vehicle (typically 100 μL). Experiments were carried out 3-4 weeks after the last Tamoxifen (or vehicle) injection. Muscle-specific Hfe2 congenital knockout, homozygous Hfe2$^{\Delta Acta-Cre}$ transgenic mice were maintained on a C67B1/129S1 mixed background (The Jackson Laboratory, ACTA1-cre, FBV.Cg-Tg(ACTA1-cre)79Jme/J, 006139). Wild type mice were purchased from The Jackson Laboratory (000664) and mice between 6-8 weeks of age were selected for experiment. Mice were maintained on C57BL/6 J background. Genotyping of all animals was done after completion of the experiments for blinding purposes. Male and female mice were used in our analyses, with the exception of EAE studies that only used female mice. List the primers used for genotyping of animals with RTq-PCR, can be foung in Supplementary Table 1.

### Ex-vivo light-sheet imaging & image analysis

Mice were injected with Lysine-fixable Texas Red-conjugated 70 kDa dextran (TR-dextran) (D1864, Thermo Fisher) via tail vein and left undisturbed for one hour to allow enough TR-dextran circulation duration. Mice were subsequently anesthetized with isoflurane and transcardially perfused with 15 mL of phosphate buffer saline (PBS) followed by 5 mL of 4% paraformaldehyde (PFA). Brains were collected and submerged in 4% PFA overnight. Brain tissues were lipid-cleared using advanced CUBIC protocols[53]. Samples were incubated in 50 ml of CUBIC-1 solution containing a mixture of urea (25 wt%, Bioshop, URE001.500), Quadrol (25 wt%, Sigma, 22262-1 L), Triton X-100 (15 wt %, Bioshop, TRX777.500) and ddH$_2$O (10 wt%) at 37 °C for 7 days. CUBIC solution was replaced daily. Samples were stored in CUBIC reagent at room temperature upon complete optical transparency of the tissue. Optical-transparent samples were imaged on a commercially available light-sheet fluorescence microscope (LSFM) (UltramicroscopeII, LaVision BioTec) equipped with an optimized Zoom Body (MVX-10, Olympus, Japan), a sCMOS camera (Andor Neo5.5) and a 2× objective lens with a 6 mm working distance dipping cap (Olympus). Images were taken at 590 nm excitation wavelength at 30% laser power with 2.5× zoom. Light-sheet microscopy images were acquired at 2.2 Hz using 250 ms exposure with each frame offset by 10 μm steps across the half-brain volume. Light-sheet images were gamma corrected for display purposes. Maximum intensity projections of stack images were performed using ImageJ (NIH, https://imagej.nih.gov/ij/). The 3D reconstructions and manipulations were done using the Imaris9.0 surface module (Bitplane, http://www.bitplane.com/imaris). Analysis was done by normalizing total fluorescence intensity to the total brain volume.

### In-vivo multiphoton imaging & image analysis

Male and female mice were first anesthetized by intra-peritoneal injection of avertin at a dose of 250 mg/kg. For local analgesia 100 μL of bupivacaine (2.5 mg/ml) was injected into the scalp. A craniotomy was performed with dental drill to create an optical window over the parietal cortex. Cranial window was made with 0.5% agarose covered with 8 mm diameter round cover slip. Mice were subsequently received 150 μg of TR-dextran (D1864, Thermo Fisher) at a concentration of 10 mg/mL via tail vein. Cerebral vessel was imaged using a Leica two-photon microscope equipped with 25× objective water immersion lenses. Texas Red fluorophore was excited at 1100 nm with 5% laser power and emission was detected with 600-800 nm filters. The field of view (348 μm x 348 μm) was chosen pseudo-randomly such that several vessel types were observed. A vertical range of approximately 200 μm, which included the cortical surface near the top, was selected for axial scanning in 10 μm steps. We chose a scan speed of 700 [a.u.], which corresponded to a 1.5 Hz frame rate or approximately two full volume scans per minute. Detector gain was adjusted automatically to avoid saturated pixels. Time-lapse images were continuously taken for 5 to 45 minutes. All statistical analysis was done using ImageJ. The acquired z-stacks at each time-point were projected to a single image based on

the maximum sample value. In separate controls, the dura was imaged with second-harmonic generation imaging of collagen, which is absent inside the brain and precisely defines the boundary between the central nervous system and the rest of the body. Diffuse dye leakage observed above the most superficial blood vessels was discarded before applying the maximum value projection. These single projected images were normalized to the 5 min time-point using standard ΔF/F normalization method with prior subtraction of tissue background at each time-point. Arterioles and venules were excluded from the mean fluorescence intensity computations for each time-point. When local leaks were observed superficial to an excluded vessel, typically overlapping the vessel edge, the region of vessel exclusion was adjusted frame-wise to include this leakage signal. This analysis was facilitated by a simple semi-automated vessel segmentation plugin we internally developed for ImageJ. Pooled normalized time series were compared among different groups at each time points. To assess overall differences between groups we compensated for the initial leakage before the 5 min time-point using a more absolute measure of leakage: the fraction of tissue region pixels above the mode vessel region intensity. Groups were pairwise contrasted using repeated measures ANOVA applied to the pool pixel leakage fraction after log-rescaling to correct for positive skew. Graphically, these were represented as the time averaged mean and sem.

### AAV8 viral rescue experiment

Deletion of liver Hfe2 production in adult mice was achieved by injecting 6-8 weeks of Hfe2 fl/fl via jugular vein with an average dose of $5 \times 10^{11}$ genome copies of either AAV8-AlbCre-TBG (VB1724) or AAV8-GFP (VB1743) viruses (Vector Biosystems Inc). One week after injections, controls were injected with PBS while mice injected with AAV8-AlbCre- or AAV8-TBGCre were injected via IP injection with either 20ug Hfe2 or 40ug 4ig protein purified from HEK cells. Mice were injected with these proteins once a week for a total of 4 weeks. After 4 weeks, mice were injected with Lysin-fixable Texas Red-conjugated 10 kDa dextran (ThermoFisher) via tail vein and left undisturbed for one hour to allow the dextran to circulate. Mice were then anaesthetized with isofluorane and transcardially perfused with 15 mL PBS followed by 5 mL of 4% PFA. Brains were collected and submerged in 4% PFA overnight. Brain tissues were lipid-cleared using the iDISCO clearing protocol[54]. All tissues were kept away from light during the entire clearing process. Brain tissues were first subjected to dehydration by placing in 20%,40%, and 80% methanol series for one hour in each solution and then 100% methanol overnight at RT. The next day, the dehydrated brain tissue was placed in a solution with 66% dichloromethane (DCM) (SIGMA) and 34% methanol for 1 hour and then transferred to 100% DCM solution for 15 minutes at RT. The DCM was replaced with dibenzyl ether (DBE) solution and stored at RT for 2 hours to allow for clearing. Tissues were then imaged under light sheet fluorescent microscopy as previously described.

### Immunohistochemistry

Animals were perfused with 20 mL of PBS, followed by 10 ml of 4% PFA. Animal tissues were collected and incubated in 4% PFA overnight in 4 °C. After overnight fixation, tissues were washed 3× with PBS and left to sink in the 30% sucrose / PBS solution in 4 °C for five days. Tissues were then embedded into the 30% sucrose / Optical Cutting Temperature (OCT) compound (Tissue-Tek, VWR) in a 1:1 ratio and placed on dry ice until frozen. The frozen OCT-embedded brains were stored at −80 °C and 12-14 μm cryosections were mounted on slides. The slides were then stored in −80 °C and prior to staining were pre-warmed at room temperature for 20 minutes. The slides were re-hydrated with PBS for 5 minutes, before a 5 min permeabilization step in 0.1% Triton X-100 in PBS (PBST). Sections were then blocked with 5% bovine serum albumin (BSA) in PBS for one hour. After blocking, slides were incubated with the primary antibody diluted in 1% BSA in PBST

overnight at 4 °C (antibody list below). The slides were then washed in PBST before incubation with the species-appropriate secondary antibody at a dilution of 1:1000 in PBS + 10% FBS + 0.3% triton for 1 hour at room temperature. The slides are washed twice with 0.3% PBST, with a 5 min DAPI stain in between. Following the final wash, coverslips were mounted using DAKO mounting medium (Agilent). Slides were then stored in the dark at 4 °C until imaging. All antibodies are listed in supplementary Table 1. Cell counting of NeuN stained cryosections were manually done with the ImageJ.

## Western blot
Equal amount of proteins was loaded on 10% SDS-PAGE gel and separated by electrophoresis. The gel was transferred onto nitrocellulose membrane. Membrane was blocked with 5% BSA in PBS for one hour and incubated in primary antibodies diluted in 1% BSA overnight at 4 °C. The following day, cells were washed with PBST (0.1% Tween) and incubated with species-appropriate secondary antibodies for one hour. Membrane was imaged using Odyssey® Classic Blot Imager (Li-COR). Relative abundance of the protein was normalized to the loading control using ImageJ. All antibodies are listed in Table 1. Prior probing for Hfe2/RGMa in serum, albumin was removed using the Thermo-fisher albumin depletion kit following manufacturer instructions (Cat#85160).

## Confocal imaging & image analysis
Stained and mounted tissue sections were imaged using Zeiss confocal laser scanning microscope 780 equipped with 40× water immersion objective. Specific fluorophore was excited at either 405, 488, 561 or 633 nm excitation wavelength. Pinhole was consistently set at 1.00 Airy units. The z-stack images were captured with 8–10% laser power and digital gain of 600-700 [a.u.]. Quantitative analysis for fibrinogen deposition was done by measuring fibrinogen fluorescence intensity in the perivascular space (outside isolectin staining of vasculature)[22]. Imaging was performed on images taken at the same intensity.

Immune cell infiltration was measured by normalizing total fluorescence intensity of immune cells to the total tissue area. For confocal TR-dextran imaging, 50 μm coronal brain sections from vibratome were incubated in blocking/permeabilization solution (1% bovine serum albumin, 0.5% TritonX-100 in PBS) overnight at 4 °C, followed by incubation in conjugated antibody solution overnight at 4 °C. Sections were mounted in Prolong Diamond antifade reagent. All analysis was done using ImageJ.

## ELISA
Mice blood sample was collected by cardiac puncture. Collected blood was left to clot in room temperature for 1 hour and centrifuged at 8000 rpm for 30 mins. Top layer serum was collected and stored in −20 °C. Mouse RGMa and Hfe2 ELISA assay was done using Quantikine® ELISA kit (R&D, MRGMAO and MRGMCO, respectively). All serum concentration was measured using Pierce™ BCA Protein Assay Kit. The optical density of ELISA plate was read using microplate reader (BioTek EL311 AutoReader) at 450 nm wavelength. Results were calculated based on standard curve.

## Competitive binding assay
A 96-well microtiter plate (Corning) was coated with 100 μL (10 μg/mL) of Poly-L-Lysine (Sigma, P8920) at 4 °C overnight. Wells were then washed with 100 μL of PBST (+0.02% Tween-20). 50 μl (2.5 μg/mL) of purified ligands (Hfe2 and RGMa) were then coated onto each well for 1 hr at 37 °C. Each well was then blocked with 300 μL of 3% BSA in PBST for 1 hr at 37 °C. Following blocking, 50 μL (1.0 μg/mL) Alkaline Phosphatase (AP)-tagged Neogenin with various ligand combination in 1% BSA + PBST were added to each well and incubated at 37 °C for 1 hr. Each well was washed thoroughly with 100 μL PBST and equilibrated using AP developing buffer (100 mM NaHCO3, 1 mM MgCl2). The

reaction was developed using AP developing buffer supplemented with, p-nitrophenyl phosphate (pNPP, Sigma-Aldrich). The reaction was terminated with 50 μL (0.1 M) NaOH. The absorbance of each reaction was measured using a microplate autoreader (BioTek EL311 AutoReader) at 405 nm wavelength.

## Endothelial cell isolation & membrane extraction
To extract endothelial cells, ten mice brains were collected and homogenized together with Dounce tissue grinder in ice cold MCDB131 media (Dibco, 1935552). Homogenized tissue was resuspended in 15% 70 kDa dextran (Sigma, 44886) and centrifuge at 10,000 g for 30 mins. The pellet was digested in 1 mg/mL collagenase & dispase (Roche, Colldisp-ro) in 37 °C for 2 hours. The digested product was resuspended in 45% Percoll and centrifuged at 20,000 g for 10 mins. Isolated endothelial cells at the top layer were collected and washed with ice cold PBS. Isolated cells were then place in membrane HB buffer (10 mM HEPES, 25 mM KCl, 5 mM MgCl₂) and resuspended thoroughly. Sucrose gradient was created using 50% sucrose overly with 5% sucrose. Resuspended cells in HB buffer was added on top of 5% sucrose and centrifuged at 28,000 rpm for 10 mins. Membrane in the middle phase was removed and concentration was measured using a Pierce™ BCA Protein Assay Kit. Equal amount of membrane was used for Western blot.

## Isolation and culture of human primary endothelial cells
The human primary endothelial cells used for our in-vitro experiments were isolated from freshly resected tissue from temporal lobectomy specimens obtained from the main study group from Toronto Western Hospital human primary endothelial cell (REB#13-6009). Brains from both male and female donors were collected. The following protocol was retrieved from Nikolaev et al. 2018[55]. Once received, tissue samples were rinsed with PBS, cut into approximately 3mm² fragments, and then exposed to 0.1% collagenase (Sigma) at 37 °C for 20 minutes. The tissue was then mechanically disintegrated using a 2 mL pipette and subsequently filtered through a 100 μm cell strainer (BD Biosciences). The resultant cell suspension was subjected to centrifugation at 2000 RPM for 5 minutes. After centrifugation, the cells were washed and resuspended in EBM-2 media (Lonza) supplemented with EGM-2 MV SingleQuots (Lonza). The endothelial cells were isolated using anti-CD31 Dynabeads (Life Technologies) according to the manufacturer's guidelines. CD31-positive and CD31-negative cell fractions were cultured in endothelial (EGM-2, Lonza). Cells were fixed with 4% PFA, permeabilized with 0.5% Triton-X-100, and blocked with 5% normal goat serum for immunohistochemistry. Rabbit polyclonal CD31 antibody (ab28364, Abcam, 1:20) was applied to the cells for 1.5 hours at RT. After three rounds of washing, cells were subjected to a 1-hour incubation with goat anti-rabbit IgG, Alexa Fluor 568 secondary antibody (A11011, Thermo Fisher, 1:500) at RT. Coverslips were mounted in DAPI-containing mounting media and imaging was performed using a Zeiss AxioImager fluorescence microscope.

## Transwell brain endothelial monolayer permeability assay
The endothelial monolayer permeability assay was performed as previously described DiStefano et al.[56] Murine Brain Endothelial cells (bEnd3) (ATCC®CRL-2299™) were seeded on 3 μm pore Transwell inserts (Corning Life Sciences) coated with 10 μg/ml human plasma fibronectin (Sigma, FC010). The cells were grown on the inserts for two days until confluence. The cells were incubated in media containing 0.1% FBS and treated overnight with 5 μg/mL purified RGMa with or without 10 μg/mL Hfe2. TNFα (10 ng/ml) was added as a positive control. After treatments, horseradish peroxidase (HRP, Sigma, P8375) was added into the upper chamber at a concentration of 1.5 μg/ml and tracer flux was allowed for 2 hours. The HRP accumulation in the bottom chamber was assessed using a TMB colorimetric assay (Cell Signaling). 10 μL aliquots of media from the lower chamber were

## Table 1 | Antibodies

| Primary Antibody | Species | Vendor, Catalog | RRID | Dilution |
|---|---|---|---|---|
| Hfe2 | Goat, polyclonal | R&D, AF3720 | AB_2264104 | 1:200 |
| CD31 | Goat, polyclonal | R&D, AF3628 | AB_2161028 | 1:100 |
| Neogenin | Rabbit, polyclonal | Santa Cruz, H-175 | AB_10609239 | 1:100 |
| Neogenin | Goat, polyclonal | Santa Cruz, C-20 | AB_630839 | 1:100 |
| Isolectin | Griffonia simplicifolia | Invitrogen, I21411 | AB_2314662 | 1:200 |
| Fibrinogen | Rabbit, polyclonal | Abcam, ab34269 | AB_732367 | 1:200 |
| RGMa | Goat, polyclonal | Santa Cruz, Y-13 | AB_2269554 | 1:100 |
| RGMa | Goat, polyclonal | R&D, AF2459 | AB_355273 | 1:200 |
| Transferrin | Mouse, monoclonal | Santa Cruz, F-8 | AB_2077957 | 1:1000 |
| CD3 | Hamster, monoclonal | BioLegend, 145-2C11 | AB_312666 | 1:100 |
| B220 | Rat, monoclonal | BioLegend, RA3-6B2 | AB_10933424 | 1:200 |
| CD11b | Rat, monoclonal | BioLegend, M1/70 | AB_312784 | 1:200 |
| CD4 | Rat, monoclonal | BioLegend, GK1.5 | AB_312686 | 1:200 |
| Claudin-5 | Rabbit, polyclonal | Abcam, ab15106 | AB_301652 | 1:300 |
| GAPHD | Mouse, monoclonal | Abcam, ab8245 | AB_2107448 | 1:500 |
| CD13 | Goat, polyclonal | R&D, AF2335 | AB_2227288 | 1:250 |
| HIF1a | Rabbit, polyclonal | Invitrogen, 16H4L13 | | 1:1000 |
| Occludin | Rabbit, polyclonal | Invitrogen, 71-1500 | AB_88065 | 1:1000 |
| PLVAP | Rabbit, polyclonal | Cell signalling, 38238 | | 1:1000 |
| YAP | Rabbit, polyclonal | Cell signalling, 4912 | | 1:1000 |
| P-AKT | Rabbit, polyclonal | Invitrogen, 44-625 | | 1:1000 |
| pSMAD1/5/9 | Rabbit, polyclonal | Cell Signaling, 13820 | | 1:1000 |
| **Secondary Antibody** | **Fluorophore** | **Vendor, Catalog** | **RRID** | **Dilution** |
| Donkey anti-goat IgG | Alexa Fluor 488 | Invitrogen, A11055 | AB_2534102 | 1:1000 |
| Donkey anti-goat IgG | Alexa Fluor 594 | Invitrogen, A11058 | AB_2534105 | 1:1000 |
| Donkey anti-goat IgG | Alexa fluor 647 | Invitrogen, A-21447 | AB_141844 | 1:500 |
| Donkey anti-rabbit IgG | Alexa Fluor 594 | Invitrogen, A21207 | AB_141637 | 1:1000 |
| Donkey anti-rabbit IgG | Alexa Fluor 488 | Invitrogen, A21206 | AB_2556546 | 1:1000 |
| Goat anti-mouse IgG | IRDye 800 | LI-COR, 925-32210 | AB_2687825 | 1:10,000 |
| Goat anti-rabbit IgG | IRDye 680 | LI-COR, 925-68071 | AB_10956166 | 1:10,000 |
| Donkey anti-rat IgG | Alexa Fluor 488 | Invitrogen, A21208 | AB_2535794 | 1:1000 |
| Donkey anti-rat IgG | Alexa Fluor 594 | Invitrogen, A21209 | AB_2535795 | 1:1000 |

placed in a 96-well plate in triplicate and 100 μL of TMB was added. The reaction was allowed to proceed for one minute alongside a standard curve of HRP to empirically determine tracer concentration in the lower chamber. Then 100 μL of 1 N HCL was added to terminate the reaction and absorbance was read at 450 nm. Raw absorbance values were converted to concentrations using the HRP standard curve.

Additionally, to assess leakage using molecules like those used in vivo, 70 kDa FITC-dextran was used in complementary bEnd3 and human primary endothelial cell (REB#13-6009) transwell assays instead of HRP. After treatment, media was replaced with Hank's Balanced Salt Solution (HBSS) with 1 mg/mL 70 kDa FITC-dextran in the transwell inserts for 30 minutes. Then 100 μL aliquots from the lower chamber were placed in a 96-well plate in triplicate and absorbance was measured at 490 nm. Raw absorbance values were converted to concentrations using the FITC standard curve.

### Trans-endothelial electrical resistance measurements (TEER)
bEnd3 cells were first thawed and plated onto Cellstar® 6 cm tissue culture dish (Greiner Bio-One, 628160). Cells were cultured with 89% DMEM (Sigma, D5796-500ml), 10% FBS (Gibco, 12483-020) and 1% Pen Strep containing 100 U/mL penicillin and 100 μg/mL of streptomycin (Gibco, 15140-122). bEnd3 were maintained in a humidified incubator at 37 °C and 5% $CO_2$ environment. Upon confluency, bEnd3 were seeded into 24-well polyester membrane Transwell® inserts (0.4 μm pore diameter) (Corning, 3470) with a

seeding density of 75,000 cells/inserts. TEER values were measured daily using Endohm-6 voltohmmeter chamber (World Precision Instruments) and resistance measurement reading was computed by EVOM2 (World Precision Instruments). Blank resistant value was subtracted from total resistance and further multiplied by the total surface area of the insert to obtain final TEER values ($\Omega cm^2$). Ligand treatments were administered at 1 μg/insert into the culture media. TEER values were monitored in 5, 60, 120, and 180 min time intervals.

### Immunocytochemistry & cell lysate
24-wells round coverslip bottom were coated with 100 μl (10 μg/mL) of Poly-L-Lysine at 4 °C overnight. bEnd3 cells were seeded directly on top and maintained in a humidified incubator at 37 °C and 5% $CO_2$ environment. Upon 90% confluency, bEnd3 was treated with various ligand combination overnight. For immunostaining, cells were fixed with 4% PFA and blocked with 5% BSA in PBS for one hour. Cells were then incubated in primary antibodies diluted in 1% BSA overnight at 4 °C. The following day, cells were washed with PBS and incubated with species-appropriate secondary antibodies for one hour and covered with coverslip using Dako mounting medium. For western blotting, cells were washed with ice cold PBS and lysed in 1x RIPA buffer (Cell Signaling). The lysates were scraped into Eppendorf tubes and centrifuged at 17000 g for 10 mins. Protein concentration was measured using Pierce™ BCA Protein Assay Kit. Equal amounts of cell lysate were

mixed with 2x Laemelli buffer and boiled at 95 °C for 10 mins. All antibodies are listed in supplementary table 1.

### PDGF-B expression in vitro and in vivo

Culture bEnd3 cells were treated with RGMa and Hfe2 along with VEGF. Cells were then stained with PDGF-B antibody after 24 hours. RT-qPCR was used to quantify PDGF-B gene expression level in RGMa and Hfe2 treated bEnd3 cells. Cells were harvested after 24 hours and expression was normalized to GAPDH housekeeping gene. In vivo, PDGF-B expression was assessed from isolated brain endothelial cells in wild type and Hfe2$^{\Delta alb\text{-}cre}$ mice. Endothelial cell isolation procedure described above.

### Quantitative RT-qPCR

Sorted endothelial cells (from FACS isolation) were lysed and RNA was extracted via RNeasy Plus Mini Kit (Qiagen). Extracted RNA was digested with RQ1 RNase-free DNase (Promega, Ref #M6101) to prevent DNA contamination. Digested RNA sample was reverse transcribed and Quantitative PCR was performed with SYBR green master mix using Luna® Universal One-Step RT-qPCR Kit (NEB, #E3005) with 200 nM of primers. The reaction mixture was loaded into the Mastercycler Eppendorf Realplex for running 40 cycles followed by a melting curve analysis to confirm the purity of the products. Gene expression was determined using the comparative CT method with mRNA levels normalized to control housekeeping gene, GAPDH.

### Induction and scoring of Experimental Autoimmune Encephalomyelitis (EAE) mice

Female mice were subcutaneously injected with an emulsion of 50 µg myelin oligodendrocyte glycoprotein (MOG) amino acids 35-55 (Sheldon Biotech, Montreal, QC) peptide in incomplete Freund's adjuvant (Sigma) supplemented with 1 mg of mycobacterium tuberculosis (CFA) (Difco). 400 ng of pertussis toxin (List Biologicals) was administered intraperitoneally on days 0 and 2 post-immunization. For ligand treatment, 40 µg of purified protein was administered intravenously once every three days for the duration of the experiments. Disease severity was assessed using standard body conditioning scores: 0, no paralysis; 1, loss of tail-tone reflex; 2, loss of righting reflex; 3, complete hind limb paralysis; 4, forelimb weakness; 5, moribund or dead. Intermediate scores (.5) were given for animals which did not meet the upper scale of paralysis. A mean cumulative score was obtained from two assessments per day at 12-hr intervals.

### Adoptive transfer EAE protocols

Donor WT mice were immunized with MOG and CFA. Animals were sacrificed 10 days post-immunization. Splenocytes and lymphocytes were isolated and re-stimulated with 50 ug/ml of MOG peptide and 10 ng/ml recombinant mouse IL-23, 10 ng/ml recombinant mouse IL-1α, 25 ug/ml anti-IFNγ, and 10 ug/ml anti-IL-4 (Biolegend) for 72 hours. After culture, CD4 T cells were isolated using a Miltenyi CD4 T cell isolation kit, stained with CellVue Maroon (Affymetrix) and 4 to 5 million cells were injected into the recipient wild type mice. Recipient mice were injected with pertussis toxin on days 0 and 2 and treated with 40 µg of Hfe2 once every three days. Spinal cords were collected and examined for cellular infiltrates on day 8 post-injection. The inflammatory index was based on the inflammatory index of H&E staining. Briefly, 0 = no cells. 1 and 2 had infiltration in the parenchyma of the spinal cord with 1 = 1-10 cells and 2 = 10 or more cells. 3 and 4 had infiltration within the white matter and grey matter with 3 = 1-10 cells and 4 = 10 or more cells. 4 sections from each animal were scored and an average calculated per animal.

### Morris Water Maze (MWM)

Mice received visible-platform training for 3 days (4 trials per day) and hidden platform training for 12 days (4 trials per day). If the mice did not find the platform within 90 s, then they were guided to the platform by the experimenter's hand. For hidden-platform training, the platform was submerged under 1.5 cm of water; the procedure is otherwise the same as that used for visible-platform training. The location of the platform was changed in hidden-platform training compared to visible-platform training. For reversal training, the location of the platform was changed to a quadrant different from that used for regular hidden-platform training. On days 6, 9, 12, and 15, mice were given a 60-s probe trial, where they were allowed to explore the maze without a platform. The intertrial interval was roughly 10–15 min. Learning was assessed by evaluating the time and distance required to find the hidden platform in the training trials, and memory was measured by examining the time spent during the probe trials in the quadrant of the pool where the platform was previously located.

### Open Field Test

Mice were placed into walled apparatus with sensors attached to record their position continuously for 1 hour. Positions and breaks in beam sensors were recorded and analyzed by Amonlite-Activity Monitor Software. Data output was in total activity, fast activity, slow activity, active time, static time, distance travelled in counts of beam breaks per time.

### Marble burying test

Cages were half filled with wood shaving bedding, and 20 marbles were lined approximately 2 cm apart within the cage. Mice were each placed in the cage and left to freely dig around for an hour. At the end of the session, marbles which were covered at least two-third with bedding were counted as buried.

### Pericyte coverage immunohistochemistry

Sections were blocked with 5% normal donkey serum (Vector Laboratories)/0.1%Triton-X/0.01 M PBS for 1 h and incubated with polyclonal goat anti-mouse aminopeptidase N/CD13 for pericyte coverage. After incubation with the primary antibody, sections were washed in PBS and incubated with Alexa fluor 647-conjugated donkey anti-goat. To visualize brain endothelial vascular profiles sections were incubated with Dylight 594-conjugated Lycopersicon esculentum lectin (Vector Labs, DL-1177; 1:200) for 1 h. For double staining of lectin and CD13, Dylight 594-lectin was incubated simultaneously with Alexa fluor 647-conjugated donkey anti-goat secondary antibody for CD13. All incubations were performed in dark to prevent fading of fluorescence. All images were taken with a BZ-9000 fluorescent microscope (Keyence Corp). Z-stack projections, pseudo-coloring and image analysis were performed using ImageJ software (US National Institutes of Health). Gain, digital offset, and laser intensity were kept standardized.

### Pericyte coverage quantification

Ten-micron maximum projection z-stacks (field area 640 × 480 µm) were reconstructed, and the areas occupied by CD13-positive (pericyte) and lectin-positive (endothelium) fluorescent signals on vessels ≤ 6 µm in diameter were subjected separately to threshold processing and analyzed using ImageJ. First, black and white 8-bit images for CD13 and lectin signals were thresholded separately using Otsu's thresholding plugin that minimize the intra-class variance of the thresholded black and white pixels, as we previously described[57,58]. After thresholding, the integrated signal density for each thresholded image was calculated. In order to express the integrated signal density as the area of the image (in pixels) occupied by the fluorescent signal, the integrated signal density was divided by 255 (the maximum pixel intensity for an 8-bit image). The integrated pixel-based area ratios of CD13 and lectin fluorescent signals were used to determine pericyte coverage as a percentage (%) of CD13-positive surface area covering lectin-positive

endothelial capillary surface area per field, as previously reported[58]. In each animal, 4–6 randomly selected fields in the cortex were analyzed in 4 non-adjacent sections ( ~ 100 μm apart), and averaged per mouse.

## Reporting summary

Further information on research design is available in the Nature Portfolio Reporting Summary linked to this article.

## Data availability

Source data are provided with this paper. The data from all figure grafts and exact p-values are provided in the Source Data file. Multiphoton Movies and light sheet volumetric data are available on request. All other imaging data are available upon request. Source data are provided with this paper.

## Code availability

All code used in the analysis of data will be made available upon request.

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

## Acknowledgements
We thank Dr. Marc Schwab, Dr. Ivan Radovanovic, and Dr. Sandra Vetiska for assistance with the endothelial cell isolation procedures. This work was supported by the Krembil Foundation, Heart and Stroke Foundation of Ontario (Grant number NA7067), The Canadian Foundation for Innovation, the Canadian Institutes for Health Research (Grant number PJT 162094), and anonymous donor, and the Vision Research Science Program of the University of Toronto for XFW and MS.

## Author contributions
Research planning and supervision were PPM; light sheet and 2 P experiments were performed by XFW and MS. Immune cell activation experiments and interpretation were done by YB and JW. Behavioral assessments were done by XFW and SL. ELISA assays were done and supervised by RV and BKM. EAE experiments and analyses were performed by RV, YB and APS, staining of human MS section and quantification were done by HR. SD and PLC performed and analyzed leakage studies using bright field imaging. Isolation of endothelial cells were done with TW. Blinded analysis of 2 P and light sheet experiments were done by DR and XFW, respectively, and both HH and JC did immuno-histochemistry. APS performed the analysis of blood vessel leakage in various organs, Transwell assay was done and analyzed by PVD and JEF, generation of Neo animal model was done by JFC. Pericyte studies and their analyses were performed by AMN and BVZ; and the manuscript was written by PPM.

## Competing interests
The authors declare no competing interests.

## Additional information

[1]Krembil Research Institute, University Health Network, Krembil Discovery Tower, 60 Leonard St., Toronto M5T 2O8 ON, Canada. [2]Institute of Biomedical and Biomaterial Engineering, University of Toronto, 1 King's College circle, Toronto M5S 1A8 ON, Canada. [3]Department of Physiology, Faculty of Medicine, University of Toronto, 1 King's College circle, Toronto M5S 1A8 ON, Canada. [4]Department of Physiology and Neuroscience, The Zilkha Neurogenetic Institute,

Keck School of Medicine of the University of Southern California, Los Angeles, CA, USA. [5]University of Maryland, School of Medicine, Department of Neurology, Baltimore, MD 21201, USA. [6]Toronto General Hospital Research Institute, University Health Network, 101 College St. Rm 3-308 Toronto M5L 1L7 ON, Canada. [7]BioNTech resano GmbH, An der Goldgrube 12, 55131 Mainz, Germany. [8]Group of CNS Angiogenesis and Neurovascular Link, and Physician-Scientist Program, Institute for Regenerative Medicine, Neuroscience Center Zurich, and Division of Neurosurgery, University and University Hospital Zurich, Zurich, Switzerland. [9]Division of Neurosurgery, University Hospital Zurich, Zurich, Switzerland. [10]Division of Neurosurgery, Department of Surgery, Toronto Western Hospital, University Health Network, Toronto, Canada. [11]The Neuro - Montreal Neurological Institute and Hospital, 3801 Rue Université Montréal, QC H3A 2B4, Canada. [12]Department of Ophthalmology and Vision Sciences, Faculty of Medicine, University of Toronto, 340 College St., ON Toronto M5T 3A9, Canada. [13]These authors contributed equally: Xue Fan Wang, Robin Vigouroux, Michal Syonov. ✉e-mail: Philippe.Monnier@uhn.ca

