## [Peer Review File · Nature Communications]

REVIEWER COMMENTS

Reviewer #1 (Remarks to the Author):

The authors report that the peripherally-derived serum protein Hfe2 maintains BBB integrity by blocking the interaction of brain endothelial Neogenin with circulating RGMa. They show loss of liver Hfe2 induces significant BBB leakage along with neuronal loss and behavioral deficits. Additionally, they show exogenous RGMa can induce BBB leakage, but this effect is abolished by injection of Hfe2 or knockout of endothelial Neogenin. Finally, they demonstrate that knocking out liver Hfe2 exacerbates EAE severity whereas Hfe2 supplementation reduces disease burden in the model.

The experiments are well-designed with appropriate controls. The manuscript outlines an interesting role for a liver-derived protein in maintaining BBB and CNS health. However, there are some major concerns that need to be addressed prior to consideration for publication. Namely, whether these findings are specific to the brain endothelium or the BBB, and whether knock-out of the Neogenin receptor on brain ECs rescues the effects of Hfe2 knock-out. Additional major and minor comments as outlined below.

Major Comments:

The title claims specificity to the brain vasculature. However, the data presented does not demonstrate whether the Hfe2 signaling pathway is indeed important specifically at the BBB. Rather, it might be generally important for endothelial cell function. Is Neo1 enriched in brain ECs compared to peripheral ECs? Or, alternatively, does Hfe2 generally support cell health instead of a BBB-specific process? Please include studies of peripheral vessel leakage to back up the claim that these effects are specific to the brain and not a global mechanism affecting all ECs.

Similarly, figure 1 and the corresponding supplemental videos demonstrate leakage in larger vessels, not confined to the small capillaries which are the site of the BBB. Is the mechanism influencing all brain ECs, rather than something specific to the blood-brain barrier? The data would suggest this is not BBB-specific, in which case the title and conclusions need to be appropriately modified throughout.

If the hypothesized role of Hfe2 is that it restores BBB permeability by preventing RGMa binding to Neogenin, then EC-specific knockout of Neogenin should rescue the vessel leakage observed in the Alb-cre Hfe2 f/f knock-out animals. This experiment seems central to the author's conclusions, since in the animal model it is not entirely clear that Hfe2-Neogenin interaction at the BBB alone is accounting for perturbations at the brain vasculature, rather than other consequences of altering liver Hfe2 production. An siRNA knockdown of Hfe2 in the Neo1-Tie2CreER mice could be a feasible strategy to test this hypothesis.

Fig 2 – the behavior looks significantly worse in KOs; is this a CNS defect and not a general sickness behavior? Does Hfe2 have other important roles in the body?

Fig 3d - The disruption of cldn5 localization is visually obvious but needs quantification.

Fig 5 - Is Neogenin expressed on immune cells? That is an important consideration to rule out direct influence on effector immune cells especially with the choice of Tie2CreER which drives Cre expression in both endothelial and hematopoietic cell lines.

The discussion is extremely brief. What do the authors speculate about the molecular pathway in ECs that is regulating the permeability? Does it converge on TJs, or maybe is it general EC/brain health? Is the dramatic loss of neurons after Hfe2 loss attributable to the BBB leakage or some other process? Is the protective effect of Hfe2 specific to MS or might it generalize to other CNS diseases?

Please remove all instances of "data not shown." Either include the data in the supplement or remove these unsubstantiated claims.

Minor Comments:

Introduction is unusually broad- consider relevant background to the manuscript rather than the history of hepatic encephalopathy (paragraph 1).

Paragraph 2- sentence 3 claiming that fibrinogen may account for most of the behavioral disturbances in MS, Alzheimer's Disease, and stroke is untrue, please revise.

Regarding Hfe2 expression in the brain- extended data figure 1b demonstrates lack of immunostaining in the brain, and a citation from 2004 on absent expression. There have been several brain transcriptomic atlases in humans and mice since that publication- is Hfe2 similarly not expressed in the brain in the more recent works?

The authors describe the liver-brain axis in the introduction in reference to the work in this manuscript. Is there evidence that Hfe is dysregulated in hepatic encephalopathy? How is liver-derived Hfe2 expected to be altered in CNS-specific inflammatory conditions such as MS? These deserve a more thorough explanation and discussion since the link is not entirely intuitive.

The authors mistakenly refer to AAV8 as an "adenovirus". It is an adeno-associated virus.

The authors should include a brief discussion of this recent paper whose findings are highly related to the present study: RGMA Participates in the Blood–Brain Barrier Dysfunction Through BMP/BMPR/YAP Signaling in Multiple Sclerosis <https://www.ncbi.nlm.nih.gov/pmc/articles/PMC9159795/> doi: 10.3389/fimmu.2022.861486

Reviewer #2 (Remarks to the Author):

In this manuscript Wang, Vigouroux et al. could show that Hfe2-protein has an impact on the maintenance of the blood-brain barrier. They state that Liver- and/or muscle-born Hfe2 competes with RGMA for binding to Neogenin, thereby blocking RGMA-induced BBB breakdown.

The authors provide in vitro as well as in vivo data to support their hypothesis. In more detail, they showed that conditional knock out of Hfe2 in the liver leads to a leakage of the BBB for 70 kDa TexasRed-conjugated Dextran and blood-born Fibrinogen. Furthermore, these KO-mice showed an impairment in their cognitive functions and behaviour.

The used in vitro model with bEnd3 cells showed that Hfe2 counteracts the effects of RGMa, which are a reduced expression of PDGF and disruption of Claudin-5 pattern. However, when Neogenin is depleted from endothelial cells treatment with RGMa does not lead to any leakage in the brain of an inducible KO-mouse model.

Hfe2 is widely described in literature in context of iron-dependent dysfunctions in various context like hereditary hemochromatosis. A liver- and skeletal muscle-Hfe2 knock out mouse model has been described before in context of changes in systemic iron homeostasis (DOI: 10.1002/hep.24547, DOI: 10.1016/j.cjca.2017.04.013). However, the consequences for the breakdown of the blood-brain barrier in Hfe2-knock out models are new. Still, the authors miss out on transfer these findings in mouse to human data/in a human model.

Major concerns

Hfe2 is primarily expressed in skeletal muscles and secondary in the liver. In this study Hfe2 serum levels of muscle KO mice was not as much decreased as in the liver knock out model. Nonetheless, the knockout of muscle specific Hfe2 does not impact the serum iron levels like it has already been described before (DOI: 10.1002/hep.24547). Still the impact of the muscle-specific KO can be doubted. Authors should comment on how the less effective KO of muscle-specific Hfe2 is relevant for the studies.

Some of the depicted figures are too small and therefore it is hard to see what is shown (e.g. in a staining) and if this is matching with the explained results. In most of the cases a larger size and improved quality is necessary.

o 2a, 3d, 4c, 5b, 5e, 6d

Why did the authors not check for a disruption of other markers beside Claudin-5 and Occludin?

Even though bEnd3 cells are commonly used for BBB studies the interaction of RGMa and Hfe2 would be more representative if these studies were also done with human primary BMECs or iPSC-derived BMECs.

Since the authors check for invasion of TR-Dextran and Fibrinogen in vivo, it would be interesting to see the same proteins used in the in vitro permeability transwell assays

In figure 5a the result of the Hfe2-coated ELISA with Neo-AP+Hfe2 leads to a significant decrease in the binding capacity which is even worse than with RGMa. Please comment on that.

Minor concerns

The authors should change “model of MS” to “model mimicking aspects of MS”. (DOI 10.1007/s00401-016-1631-4) Furthermore, authors should highlight other possible diseases that might be influenced by this found mechanism and comment for example on an involvement of RGMa/Hfe2 in hepatic encephalopathy.

It would be helpful to make the used KO-mouse model more visual in figures

Comment on the varying sample size per group in for example figure 3c since it should be easy to have the same n-number

In line 297 Extend. Data Fig. 10 should be consistent with the way you labelled figures before

In the very short discussion the authors do not comment on the EAE model to study MS-like pathology related to Hfe2 expression

Reviewer #3 (Remarks to the Author):

Wang et al., perform comprehensive study demonstrating roles of liver and muscle secreted Hfe2-protein to maintain blood brain barrier (BBB) integrity. They utilized genetic and acute deletion model of liver and muscle Hfe2-protein, showing increased BBB leakage using tissue clearing and light sheet microscopy. They also provide convincing evidence that Hef2 prevents harmful effect of RGMa via competitive binding to Neogenin using in vitro and in vivo studies. Lastly, they present that Hef2, RGMa, and Neogenin play an important role in progression of multiple sclerosis. Overall, this is an impressive study with many carefully designed control experiments, providing convincing evidence of surprising involvement of peripherally secreted molecules to affect BBB permeability. This finding has significant clinical relevance with many peripheral and brain disorders. I do have a few suggestions to further improve the manuscript.

Major points

- Authors perform tissue clearing and 3D microscopy to image the whole brain. So, it's puzzling why they only present a part of the brain. It'd be very interesting to see whether the whole brain is uniformly affected or some areas showing more vulnerabilities. Conversely, what areas would be more resilient in this insult. Such data can help to identify potential affected neural circuits like with behavioral deficit.
- Overall, there is no mention about which parts of the brain each histology images are from. For example, Fig1c-e, Fig2a, Fig3a,d, etc, please include anatomical areas in the figure legend clearly, so readers can understand where it comes from.
- It'd be good to include why particular behavioral tests were chosen. For example, Morris water maze will rely on the hippocampus. Is the hippocampus affected by your experimental models? If so, it'll be good to include some explanation in the result section before going over the behavioral result. There is a comment on neuronal loss in the cortex. But depending on which cortex is affected, different behavior deficit can occur. Linking with affected brain regions with expected behavioral deficit will be helpful.
- Discussion can be expanded a bit further by including how body condition such as liver failure can influence brain permeability, its clinical implication, etc. Can author provide any perspective on how even normal aging can be influenced by this mechanism?

Minor comments

- What's injection route of AAV8? Tail vein? The method section needs to include the info.
- Hfe2/RGMc synonym needs to be mentioned up front.

Reviewer #4 (Remarks to the Author):

In this paper, the Authors investigate how the liver influences BBB integrity by focusing on soluble HJV. This protein is expressed mainly in the liver and skeletal muscle and originates from the plasma membrane. In the liver, HJV regulates the expression of the BMP-SMAD target gene hepcidin, thereby regulating iron metabolism. In the skeletal muscle, HJV regulates myofiber regeneration. HJV KO mice develop hemochromatosis characterized by severe body iron accumulation, myofiber atrophy, and fibrosis.

These suggestions are offered to the Authors:

1. The results reported in the first paragraph (page 4) are intriguing. However, liver-specific HJV KO develops severe iron overload. To demonstrate that iron does not play a role in BBB integrity, Hfe2fl/fl;Alb-Cre mice should be treated with soluble recombinant HJV to rescue the increased BBB leakage.
2. Page 7, line 179: As the Authors commented in the text, the results cannot exclude the functional deficit observed in Hfe2fl/fl;Acta-Cre is due to muscle weakness. To prove this is not the case, Hfe2fl/fl;Acta-Cre should be treated with recombinant HJV.
3. Since Hfe2 liver KO mice develop severe iron overload, it is unclear why the authors have performed critical experiments in this mouse model instead of using the Hfe2 sk muscle KO that does not display iron overload.
4. Soluble HJV sequesters BMPs and decreases BMP-SMAD signaling (Theurl et al., 2011): is the BMP pathway involved in BBB permeability? Is sHJV impairing BBB function by downregulating BMP-SMAD signaling?
5. The Authors hypothesize that soluble HJV, by binding to RGMa, prevents the interaction of RGMa with neogenin. Since soluble HJV binds BMPs, are these ligands necessary to mediate the RGMa- soluble HJV interaction?
6. In the EAE model, does the soluble HJV treatment also affect the inflammatory response? Is soluble HJV reduced in MS patients?
7. In skeletal muscle dystrophy, HJV is reduced (Zhang et al., 2019). Is this accompanied by increased BBB leakage?
8. Soluble HJV is not secreted since it is a GPI-anchored protein but cleaved by furin. Furin is upregulated in iron deficiency-hypoxia. Do the Authors observe if iron deficiency and/or hypoxia are also characterized by BBB leakage, likely due to increased levels of circulating soluble HJV?
9. The Discussion should be implemented: the results demonstrate that is not the liver but soluble HJV (released by the liver and the skeletal muscle) that regulates BBB integrity.

Minor:

1. page 3, line 88: the cited reference is wrong. Please check.

Re: **Wang et al, “The liver and muscle secreted Hfe2-protein maintains blood brain barrier integrity”**.

Thank you for reviewing our article as each critique stimulated new experiments and textual adjustments that improved the manuscript considerably.

This revised manuscript thoroughly addresses all the concerns raised by the reviewers with supporting data.

In view of the novelty, mechanistic depth, biological and therapeutic relevance of our extensive study, we feel that the revised paper is now appropriate for Nature Communications.

Reviewer #1:

The authors report that the peripherally-derived serum protein Hfe2 maintains BBB integrity by blocking the interaction of brain endothelial Neogenin with circulating RGMa. They show loss of liver Hfe2 induces significant BBB leakage along with neuronal loss and behavioral deficits. Additionally, they show exogenous RGMa can induce BBB leakage, but this effect is abolished by injection of Hfe2 or knockout of endothelial Neogenin. Finally, they demonstrate that knocking out liver Hfe2 exacerbates EAE severity whereas Hfe2 supplementation reduces disease burden in the model. The experiments are well-designed with appropriate controls. The manuscript outlines an interesting role for a liver-derived protein in maintaining BBB and CNS health. However, there are some major concerns that need to be addressed prior to consideration for publication. Namely, whether these findings are specific to the brain endothelium or the BBB, and whether knock-out of the Neogenin receptor on brain ECs rescues the effects of Hfe2 knock-out. Additional major and minor comments as outlined below.

Response: We thank the reviewer for the very positive and constructive comments. As described below, we have addressed all the issues raised by this reviewer.

Major: The title claims specificity to the brain vasculature. However, the data presented does not demonstrate whether the Hfe2 signaling pathway is indeed important specifically at the BBB. Rather, it might be generally important for endothelial cell function. Is Neo1 enriched in brain ECs compared to peripheral ECs? Or, alternatively, does Hfe2 generally support cell health instead of a BBB-specific process? Please include studies of peripheral vessel leakage to back up the claim that these effects are specific to the brain and not a global mechanism affecting all ECs.

Similarly, figure 1 and the corresponding supplemental videos demonstrate leakage in larger vessels, not confined to the small capillaries which are the site of the BBB. Is the mechanism influencing all brain ECs, rather than something specific to the blood-brain barrier? The data would suggest this is not BBB-specific, in which case the title and conclusions need to be appropriately modified throughout.

Response: The reviewer addresses 2 important issues.

The first one relates to the possibility that Hfe2 controls blood vessel integrity beyond the BBB. In fact, our data suggests that Hfe2 controls blood vessel integrity in the spinal cord, which is the Blood-spinal cord barrier. This suggests that Hfe2 regulates blood vessel integrity in the central nervous system. To address the possibility that Hfe2 regulates blood vessel integrity in other organs, we have performed Evans blue extravasation. As expected, Evans blue demonstrates increased extravasation in the brains of liver- and muscle-Hfe2 KO, confirming that the deletion of Hfe2 in these organs results in blood vessel leakage. We could not observe any significant difference of Evans blue extravasation in the liver, heart and muscles. This novel piece of data is presented in our novel Extended Figure 5.

We added the technical details to our Material and Methods part and added the following sentence in the results.

“To determine whether liver- and muscle Hfe2 were involved in blood vessel integrity in other

organs, we performed intra-venous injection of Evans Blue and assessed Evans blue

extravasation in the brain, heart, muscles, and liver of *Hfe2*^{ΔActa-} *Hfe2*^{ΔAlb-cre}, and *Hfe2*^{fl/fl} mice.

We observed increased Evans blue extravasation in the brain of *Hfe2*^{ΔActa-cre} and *Hfe2*^{ΔAlb-cre} mice, whereas we could not observe any difference between the muscle, heart and liver of KO vs control mice (**Extended Data Fig. 5**).

The second issue relates to the definition of the blood brain barrier. Many articles do not measure the size of the vessels and label any brain leakage as a blood-brain barrier breach. Although big vessels and small vessels have a similar cellular composition (endothelial cells + mural cells+ astroglial cell endfeet) some articles suggest that the BBB is a term that can only be used for smaller vessels. As a consequence, there is no term that describe leakage of larger vessels. To address the second issue, we have replaced BBB by blood CNS- barrier (BCB). We use this term to describe all leakages in the CNS. This will also address the first comment as we show that *Hfe2* induces leakages in the spine and the brain. We hope that the reviewer will agree with this term.

If the hypothesized role of *Hfe2* is that it restores BBB permeability by preventing RGMa binding to Neogenin, then EC-specific knockout of Neogenin should rescue the vessel leakage observed in the Alb-cre *Hfe2* f/f 5 knock-out animals. This experiment seems central to the author's conclusions, since in the animal model it is not entirely clear that *Hfe2*-Neogenin interaction at the BBB alone is accounting for perturbations at the brain vasculature, rather than other consequences of altering liver *Hfe2* production. An siRNA knockdown of *Hfe2* in the Neo1-Tie2CreER mice could be a feasible strategy to test this hypothesis.

Response: To address this issue, we considered using *Hfe2* siRNAs as suggested by the reviewer. However, the literature suggest that the in vivo effect of siRNAs is rather limited in time (1-5 days), when our data suggest that *Hfe2* downregulation alters blood vessel integrity after 3 weeks silencing. As an alternative approach, we tried to generate *Hfe2*^{fl/fl}, Neo1-Tie2CreER mice with the goal knock out *Hfe2* expression with *Hfe2*^{fl/fl} with AAV8-TBG-Cre (liver specific expression) and Neo1 expression with the Tamoxifen treatment. However, after 9 months of intense breeding we could not generate mice with the desired genotype.

In previous studies, we have generated a Neogenin neutralizing peptide (4Ig), which blocks Neogenin recruitment into lipid rafts (Tassew et al., 2014). Hence, we decided to determine whether 4Ig could revert blood vessel alterations in *Hfe2*^{fl/fl} mice treated with AAV8-TBG-Cre. As expected, we observed a significant rescue of blood vessel integrity in mice treated with 4Ig when compared to controls. This piece of data is presented in our novel Extended Data Fig. 3 3. We added the following text to our manuscript:

“If soluble *Hfe2* prevents blood vessel alteration by preventing the interaction between RGMa and Neogenin, the neutralization of Neogenin should restore blood vessel integrity in *Hfe2*^{ΔAlb-cre} mice.

Hence, we investigated the role of Neogenin by performing weekly intra-peritoneal injections of the Neogenin neutralizing peptide 4Ig²⁵ in AAV8-TBG-Cre treated *Hfe2*^{fl/fl} mice (Extended Data Fig. 3). Importantly, we observed that 4Ig treatment significantly restored BCB integrity suggesting that the Hfe2 effect on endothelial cells is mediated by Neogenin.”

Please note that all quantifications were done in a blinded fashion. Our provider had used all his stock for AAV8-Alb-Cre, for this reason we used AAV8-TBG-Cre, which also induces Cre expression in liver cells. Note that we have added details for this experiment in our Material and Methods part.

Fig 2 – the behavior looks significantly worse in KOs; is this a CNS defect and not a general sickness behavior? Does Hfe2 have other important roles in the body?

Response: We thought that at least in the case of Acta-cre induced KO, that we could observe muscle weakness, however, our measurements do not show any difference between Acta-cre x *Hfe2*^{fl/fl} and *Hfe2*^{fl/fl} controls. This data is provided in our novel Extended Data Fig. 6. The reduction in the number of cortical neurons is quite impressive and could explain the behavioral difference. Also, our data shows that Hfe2 can protect axons from RGMa-induced inhibition. It is possible that Hfe2 also acts on axons as well.

Fig 3d - The disruption of cldn5 localization is visually obvious but needs quantification.

Response: To address this issue, we have quantified Cldn5 expression in mouse brain endothelial cells treated with RGMa, Hfe2 and RGMa+ Hfe2. The quantification can be found in our novel Figure Extended Data Fig. 10. Also, we have performed similar experiments on primary endothelial cells from human brain. This novel data and the corresponding quantification can be found in our novel Fig. 3d.

Fig 5 - Is Neogenin expressed on immune cells? That is an important consideration to rule out direct influence on effector immune cells especially with the choice of Tie2CreER which drives Cre expression in both endothelial and hematopoietic cell lines.

Response: The reviewer is correct and Neogenin is expressed by immune cells. We are interested in uncovering the role of Hfe2 in EAE, and our hypothesis is that Neogenin is mediating this effect. Hence, as recommended, we investigated the impact of Hfe2 on immune cell activation. Extended data Fig 19. shows that none of the commonly used markers for immune cell activation show any difference between PBS and Hfe2 treated cells. Moreover, we performed adoptive transfer by transferring immune cells from a wt background to an Hfe2 treated animal following EAE induction in both animals (Fig. 6e). Here we observed that Hfe2 blocked cellular infiltrates, indicating that the reduction in cellular infiltrates following EAE + Hfe2 treatment is not due to an effect on immune cells but rather attributed to its impact on the blood brain barrier.

It is well accepted that restoration of the endothelial cell barrier will block immune cell infiltration following EAE. Our extravasation data suggests that blood vessel integrity is restored following Neo1 deletion, taken together our data suggests that the effects observed results from an effect on the blood barrier and not a direct effect on immune cells.

The discussion is extremely brief. What do the authors speculate about the molecular pathway in ECs that is regulating the permeability? Does it converge on TJs, or maybe is it general EC/brain health? Is the dramatic loss of neurons after Hfe2 loss attributable to the BBB leakage or some other process? Is the protective effect of Hfe2 specific to MS or might it generalize to other CNS diseases?

Response: We have considerably increased the size of our discussion, which is now over 3 pages long. Our novel discussion describes pathways and a general role of Hfe2 in brain diseases. Our data suggest that Hfe2 can neutralize the RGMa effect on axons. We are working on a study in which we are studying the role of Hfe2 injection following CNS injury.

We are also working on a manuscript in which we are addressing the role of Hfe2 in stroke. Our data shows that Hfe2 treatment prevents leakages in the CNS following middle cerebral artery occlusion (see below). We would like to have this piece of data for a manuscript that addresses the role of Hfe2 in stroke.

Fig. R1: Treatment with RGMc reduces Evans Blue penetration into CNS tissues.

Rats were subjected to MCAO followed by treatment with RGMc (70µg, tail vein injection) or PBS (Control).

a) One week after MCAO, tail vein injection of Evans Blue was performed and Brains collected (after PBS perfusion). This revealed that RGMc treatment strongly reduced (unpublished preliminary data).

Please remove all instances of “data not shown.” Either include the data in the supplement or remove these unsubstantiated claims.

Response: This has been removed or the data is now shown.

Minor Comments:

Introduction is unusually broad- consider relevant background to the manuscript rather than the history of hepatic encephalopathy (paragraph 1).

Response: We have re-written our introduction as follow:

“Clinical observations have documented that liver diseases are associated with alterations in the Central Nervous System (CNS) that manifest as behavioral changes such as cognitive dysfunction, mood disorders and sleep disturbance¹. A large body of work suggests that the accumulation of ammonia contributes to neurological pathologies following liver failure^{2,3}. A study by López-Franco et al. showed that patients with history of hepatic encephalopathy show signs of cognitive impairment even after receiving liver transplant⁴. This suggests that ammonia is not solely responsible for the alteration of brain functions observed after liver failure, and that other liver functions/factors may be involved in maintaining brain health.”

Paragraph 2- sentence 3 claiming that fibrinogen may account for most of the behavioral disturbances in MS, Alzheimer’s Disease, and stroke is untrue, please revise.

Response: This has been changed to”

“ The extravasation of toxic blood components such as Fibrinogen, may contribute to neuronal loss observed in MS, Alzheimer’s disease and stroke”

Regarding Hfe2 expression in the brain- extended data figure 1b demonstrates lack of immunostaining in the brain, and a citation from 2004 on absent expression. There have been several brain transcriptomic atlases in humans and mice since that publication- is Hfe2 similarly not expressed in the brain in the more recent works?

Response: We have looked in a few data bases and this confirms that Hfe2 is not expressed in the brain. We have added the following sentence to our manuscript:

“This was unexpected as our immunohistochemical analysis, the human protein atlas, and the Gene Set Enrichment Analysis data base of the broad institute all indicate that *Hfe2* is not expressed in the brain (**Extended Data Fig. 1d**). Therefore, this suggested that a non-brain source of Hfe2 regulates BCB integrity”

The authors describe the liver-brain axis in the introduction in reference to the work in this manuscript. Is there evidence that Hfe is dysregulated in hepatic encephalopathy? How is liver-derived Hfe2 expected to be altered in CNS-specific inflammatory conditions such as MS? These deserve a more thorough explanation and discussion since the link is not entirely intuitive.

Response: We are presenting the liver-brain axis in the introduction as this is something that is known for a long time. However, thus far, there is not much known about the factors that link the liver to the brain, this is why we thought of starting the introduction by mentioning this axis. Our data suggest that Hfe2 has no impact on immune cell activation (see Extended Data Fig. 19). Our data shows that in models for MS, the liver expresses less Hfe2.

The authors mistakenly refer to AAV8 as an “adenovirus”. It is an adeno-associated virus.

Response: This has been corrected.

The authors should include a brief discussion of this recent paper whose findings are highly related to the present study: RGMA Participates in the Blood–Brain Barrier Dysfunction Through BMP/BMPR/YAP Signaling in Multiple Sclerosis
<https://www.ncbi.nlm.nih.gov/pmc/articles/PMC9159795/> doi: 10.3389/fimmu.2022.861486

Response: The reviewer is correct and a recent study from Zhang et al., 2022, suggests that RGMA participates in blood brain barrier dysfunction. However, our data suggests that RGMA is not using YAP but another pathway to regulate blood vessel integrity. These data do not necessary contradict each other. Indeed, Zhang et al., shows that expression of RGMA in endothelial cells trigger BBB opening via YAP, while our data shows that blood borne RGMA triggers BBB opening via a YAP independent mechanism.

We have added the YAP Western Blot in our Extended Figure 13, and we have text to present this result. Also, we added the following discussion:

“ To add further evidence for a role of RGMA in MS, a recent study demonstrates that RGMA is expressed by endothelial cells and triggers BBB dysfunction via a mechanism involving YAP and BMPs³⁴. This seems to contradict our data showing that neither YAP nor the canonical BMP

pathway are involved in RGMA mediated dysfunction of blood vessels. However, the study from Zhang et al., shows that RGMA is expressed by endothelial cells and that silencing RGMA expression in endothelial cells restores blood brain barrier integrity while our study shows that blood borne RGMA is acting on endothelial cells to alter blood vessel integrity. In the study from Zhang et al., RGMA acts on endothelial cells in a cell autonomous manner³⁴, while our study involves non-cell autonomous mechanisms. This is not the first evidence that RGMA may regulate the same biological function via cell- and non-cell autonomous mechanisms²⁴. Indeed, we have shown that RGMA is expressed by growing axons and by cells surrounding these axons, and that both source of RGMA regulate axonal growth²⁴. We have identified distinct RGMA domains that regulate cell autonomous and non-cell autonomous axonal inhibition. Moreover, we showed that RGMA inhibits axonal growth via different pathways. Hence, similar to what has been observed in growing axons, our data indicate that different intracellular pathways mediate cell- and non-cell autonomous alteration of blood vessel integrity by RGMA.”

Reviewer #2 (Remarks to the Author):

In this manuscript Wang, Vigouroux et al. could show that Hfe2-protein has an impact on the maintenance of the blood-brain barrier. They state that Liver- and/or muscle-born Hfe2 competes with RGMA for binding to Neogenin, thereby blocking RGMA-induced BBB breakdown. The authors provide in vitro as well as in vivo data to support their hypothesis. In more detail, they showed that conditional knock out of Hfe2 in the liver leads to a leakage of the BBB for 70 kDa TexasRed-conjugated Dextran and blood-born Fibrinogen. Furthermore, these KO-mice showed an impairment in their cognitive functions and behaviour. The used in vitro model with bEnd3 cells showed that Hfe2 counteracts the effects of RGMA, which are a reduced expression of PDGF and disruption of Claudin-5 pattern. However, when Neogenin is depleted from endothelial cells treatment with RGMA does not lead to any leakage in the brain of an inducible KO-mouse model. Hfe2 is widely described in literature in context of iron-dependent dysfunctions in various context like hereditary hemochromatosis. A liver- and skeletal muscle-Hfe2 knock out mouse model has been described before in context of changes in systemic iron homeostasis (DOI: 10.1002/hep.24547, DOI:10.1016/j.cjca.2017.04.013). However, the consequences for the breakdown of the blood-brain barrier in Hfe2-knock out models are new. Still, the authors miss out on transfer these findings in mouse to human data/in a human model.

Response: We thank the reviewer for the very positive and constructive comments. As described below, we have obtained primary BMECs to address the lack of human data. We have also addressed all the other issues raised by this reviewer.

Major concerns

Hfe2 is primarily expressed in skeletal muscles and secondary in the liver. In this study Hfe2 serum levels of muscle KO mice was not as much decreased as in the liver knock out model. Nonetheless, the knockout of muscle specific Hfe2 does not impact the serum iron levels like it has already been described before (DOI:10.1002/hep.24547). Still the impact of the muscle-specific KO can be doubted. Authors should comment on how the less effective KO of muscle-specific Hfe2 is relevant for the studies.

Response: In these studies, we aim to demonstrate that levels of Hfe2 in the blood influences BBB integrity.

The reviewer's critique is correct, and liver derived Hfe2 plays a significant role in regulating iron levels. This is largely attributed to the presence of membrane bound Hfe2 in liver cells. To eliminate the potential influence of iron levels on blood vessel changes in our study, we examined the impact of muscle-derived Hfe2, which, following ablation, does not affect iron levels. To address the reviewer's point, we added the following text in our manuscript:

“In another attempt to evaluate the role of iron in Hfe2 KOs, we generated *Hfe2*^{ΔActa-cre} (*Hfe2*^{fl/fl},

Acta-cre) transgenic mice to genetically ablate Hfe2 production specifically in skeletal muscles.

Analysis demonstrated that this procedure reduced Hfe2 levels by $32\pm 10\%$ without altering iron levels in the serum (**Extended Data Fig. 3a-b**)¹⁵. In *Hfe2* ^{Δ Acta-cre} animals, we nonetheless observed BCB-leakages using light sheet imaging and *in-vivo* multiphoton imaging, suggesting BCB-alteration is not the result of increased iron levels (**Fig. 1f and Extended Data Fig. 1h**). Leakages appeared less pronounced than in *Hfe2* ^{Δ alb-cre} animals, which may reflect the lower decrease in blood Hfe2 in *Hfe2* ^{Δ Acta-cre} when compared to *Hfe2* ^{Δ alb-cre}. Together these results indicate that liver- and muscle-secreted Hfe2 may play a pivotal role in the maintenance of BCB-integrity. ”

Also, we decided to address this issue by performing rescue experiments, in which we treated mice with induced Hfe2 liver deletion with soluble Hfe2. This novel piece of data is presented in our novel Extended Data **Fig. 3**.

Some of the depicted figures are too small and therefore it is hard to see what is shown (e.g. in a staining) and if this is matching with the explained results. In most of the cases a larger size and improved quality is necessary. o 2a, 3d, 4c, 5b, 5e, 6d

Response: We have improved the quality of the mentioned figures.

Why did the authors not check for a disruption of other markers beside Claudin-5 and Occludin?

Response: We have checked markers such as Claudin-5, Occludin, YAP (involved in RGMa signalling), PLVAP, AKT, and HIF-1 α . We could only observe a significant difference for Claudin-5 and HIF-1 α (a mediator of BBB break-down). This novel piece of data are presented in our novel Extended Data Fig. 11-13.

We also added the following text to the result part:

“RGMa did not appear to impact the expression of Occludin, (**Extended Data Fig. 11a**).

Furthermore, we assessed the expression of a few factors that are known to trigger blood vessel dysfunction. Interestingly, we observed that HIF-1 α , a factor that regulated by hypoxia and trigger BBB opening, is upregulated in the presence of RGMa, and that addition of Hfe2 to the medium blocks this upregulation (**Extended Data Fig. 12a**). Other factors involved in blood

vessel integrity such as PLVAP²², AKT²²(**Extended Data Fig. 11b,c**).YAP²³ (**Extended Data Fig. 13**), were not influenced by the addition of RGMa to the medium. Hence our results indicated that RGMa treatment on cerebral endothelial cells significantly alters both the expression of Claudin-5, HIF-1 α and PDGF-B, which can be completely prevented by Hfe2. “

Even though bEnd3 cells are commonly used for BBB studies the interaction of RGMa and Hfe2 would be more representative if these studies were also done with human primary BMECs or iPSC-derived BMECs. Since the authors check for invasion of TR-Dextran and Fibrinogen in vivo, it would be interesting to see the same proteins used in the in vitro permeability transwell assays

Response: To address this, we have obtained human brain primary endothelial cells and tested these cells in the transwell assay. As recommended, we have used TR-Dextran to monitor permeability. We observed that human primary cells and bEnd3 cells react in a similar manner to RGMa, Hfe2, and RGMa+Hfe2 treatment. Indeed, Hfe2 prevented RGMa induced permeability in human primary cells and mouse brain endothelial cells.

We have added Material and Methods for these experiments and have added these results in our Manuscript in novel Fig. 3g

In figure 5a the result of the Hfe2-coated ELISA with Neo-AP+Hfe2 leads to a significant decrease in the binding capacity which is even worse than with RGMa. Please comment on that.

Response: The reviewer is correct, Fig.5a shows that Hfe2 leads to significant decrease. In this assay, Neo-AP interacts with the Hfe2 present on coated ELISA plates. In this assay both soluble Hfe2 and RGMa can interact with Neo-AP, thereby preventing the interaction with coated Hfe2. To address this issue, we modified the text as follow:

“To address this possible role for Hfe2, we developed an assay in which soluble Neogenin (AP-tagged) will interact with RGMa. We observed that soluble Neogenin-AP interacts with an ELISA-plate coated with RGMa (**Fig. 5a**). In a competitive binding assay, we show that Hfe2 and RGMa significantly prevented the binding of Neogenin-AP to RGMa (**Fig. 5a**). Also, we show that Neogenin-AP binds to Hfe2 and that this interaction is blocked by both RGMa and Hfe2 (**Fig. 5a**).”

Minor concerns

The authors should change “model of MS” to “model mimicking aspects of MS”. (DOI 10.1007/s00401-016-1631-4) Furthermore, authors should highlight other possible diseases that might be influenced by this found mechanism and comment for example on an involvement of RGMa/Hfe2 in hepatic encephalopathy.

Response: This has been modified as suggested.

It would be helpful to make the used KO-mouse model more visual in figures.

Response: We have added the mention of “liver KO” and “muscle KO” in the figure the first time that a specific KO is being used (Fig 1b & 1f). This should help identifying the source that is being suppressed.

Comment on the varying sample size per group in for example figure 3c since it should be easy to have the same n-number In line 297 Extend. Data Fig. 10 should be consistent with the way you labelled figures before

Response: To perform blinded studies, the genotyping of the animals is done after completion of the experiments. For this reason, the number of controls and KO may vary from one experiment to the other. We have added this detail to Methods under the ‘Animals’ section.

In the very short discussion the authors do not comment on the EAE model to study MS-like pathology related to Hfe2 expression

Response: To address this issue, we have added a whole section “ **The RGM/Neogenin pathway is involved in experimental models for MS**” to our discussion.

Reviewer #3 (Remarks to the Author):

Wang et al., perform comprehensive study demonstrating roles of liver and muscle secreted Hfe2-protein to maintain blood brain barrier (BBB) integrity. They utilized genetic and acute deletion model of liver and muscle Hfe2-protein, showing increased BBB leakage using tissue clearing and light sheet microscopy. They also provide convincing evidence that Hef2 prevents harmful effect of RGMA via competitive binding to Neogenin using in vitro and in vivo studies. Lastly, they present that Hef2, RGMA, and Neogenin play an important role in progression of multiple sclerosis. Overall, this is an impressive study with many carefully designed control experiments, providing convincing evidence of surprising involvement of peripherally secreted molecules to affect BBB permeability. This finding has significant clinical relevance with many peripheral and brain disorders. I do have a few suggestions to further improve the manuscript.

Response: We thank the reviewer for the very positive comments and the constructive review of our manuscript.

Major points

- Authors perform tissue clearing and 3D microscopy to image the whole brain. So, it's puzzling why they only present a part of the brain. It'd be very interesting to see whether the whole brain is uniformly affected or some areas showing more vulnerabilities. Conversely, what areas would be more resilient in this insult. Such data can help to identify potential affected neural circuits like with behavioral deficit.

Response: We have presented the right hemisphere of the brain because it allowed to show leakages in the subcortical area of the brain. However, we agree that a view of the complete brain will help the reader.

We have added a video of the brain in our supplementary data to show leakages of the whole brain (see Supplementary Video 5-8). We have also added images of full brain in Fig 5g for our AAV8-TBG-Cre rescue experiment and for liver and muscle KO mice in Extended Data Fig. 2 and 3.

- Overall, there is no mention about which parts of the brain each histology images are from. For example, Fig1c-e, Fig2a, Fig3a,d, etc, please include anatomical areas in the figure legend clearly, so readers can understand where it comes from.

Response: As suggested, we have added anatomical indications in the legend of our figures. For instance, Fig.1 we added:

“Representative *in-vivo* multiphoton images of TR-dextran at 40mins time-point (scale bar, 50 μ m) from the parietal lobes of the cerebral cortex.”

- It'd be good to include why particular behavioral tests were chosen. For example, Morris water maze will rely on the hippocampus. Is the hippocampus affected by your experimental models? If so, it'll be good to include some explanation in the result section

before going over the behavioral result. There is a comment on neuronal loss in the cortex. But depending on which cortex is affected, different behavior deficit can occur. Linking with affected brain regions with expected behavioral deficit will be helpful.

Response: The reviewer addresses a legitimate request and we should indicate which region fits to specific behaviors. In fact, after observing a reduction of the number of neurons in the cortex, we selected behavioral assays that could show differences when cortical neurons are affected. As such, the water maze test is mainly used to study hippocampal function, however, a few cortical areas have been shown to influence water maze learning. For instance, the retrosplenial cortex impacts water maze outcome, (see Vann, S. D., Aggleton, J. P., & Maguire, E. A. (2009)), lesions of the parietal cortex also affects water maze outcome (see DiMattia, B. V., & Kesner, R. P. 1988) also, lesions of the parietal and temporal cortex have been shown to affect water maze outcome. For this reason, we decided to use water maze evaluation.

The marble burying test and open field are normally used to study anxiety, however, a study demonstrated that these tests can be used to the effects of cortical lesions on repetitive and anxiety-related behavior in rodents (Xiao Chen et al., 2021).

We therefore mention these studies in our manuscript and added the following text to the result part:

“We decided to look at the effect of water maze which is normally used to study hippocampal functions because lesions of the parietal lobe or the retrosplenial cortex have also been shown to affect learning in this test, marble burying and open-field test are normally used to study anxiety and can be used to study the role of cortical lesions on anxiety related behavior”

- Discussion can be expanded a bit further by including how body condition such as liver failure can influence brain permeability, its clinical implication, etc. Can author provide any perspective on how even normal aging can be influenced by this mechanism?

Response: We agree that our discussion was very short. We have considerably improved our discussion, which is now over 3 pages long.

Minor comments

- What's injection route of AAV8? Tail vein? The method section needs to include the info.

Response: The injection was done using tail vein injection this has been added. Thank you.

- Hfe2/RGMc synonym needs to be mentioned up front.

Response: We agree and we are mentioning it in the abstract.

Reviewer #4 (Remarks to the Author):

In this paper, the Authors investigate how the liver influences BBB integrity by focusing on soluble HJV. This protein is expressed mainly in the liver and skeletal muscle and originates from the plasma membrane. In the liver, HJV regulates the expression of the BMP-SMAD target gene hepcidin, thereby regulating iron metabolism. In the skeletal muscle, HJV regulates myofiber regeneration. HJV KO mice develop hemochromatosis characterized by severe body iron accumulation, myofiber atrophy, and fibrosis.

Response: We thank the reviewer for his time and constructive critics. Below we address all the points raised by the reviewer.

These suggestions are offered to the Authors:

1. The results reported in the first paragraph (page 4) are intriguing. However, liver-specific HJV KO develops severe iron overload. To demonstrate that iron does not play a role in BBB integrity, Hfe2^{fl/fl};Alb-Cre mice should be treated with soluble recombinant HJV to rescue the increased BBB leakage.

Response: As requested, we have performed experiments in which mice that received liver deletion of Hfe2 were treated with soluble Hfe2. For this, we induced Hfe2 deletion in the liver with AAV8-TBG-Cre (Vector Biolabs did not have any AAV8-Alb-Cre stock any longer and their technical support indicated that AAV8-AlbCre and AAV8-TBG-Cre should have the same Cre expression in liver cells) and treated mice with a weekly injection of 20 µg of soluble Hfe2

We have added technical details for these experiments in the Material and Methods part. Also, we have added the following text to the result section:

“Hfe2 plays a major role in iron homeostasis, and *Hfe2*^{ΔAlb-cre} mice display a 52±13 % increase in iron levels in the blood (**Extended Data Fig. 4a**)¹⁵. To the role of iron levels on the BCB-leakage in *Hfe2*^{ΔAlb-cre} mice, we investigated whether soluble Hfe2 could rescue blood vessel dysfunction. We performed Hfe2 deletion in the liver by injecting AAV8-TBG-Cre in *Hfe2*^{fl/fl} mice followed by weekly tail vein injection of 20 µg of Hfe2. We observed that Hfe2 injection restored blood vessel integrity in the CNS (Extended Data Fig. 3), suggesting that iron is not involved in BCB leakage for these animals.”

2. Page 7, line 179: As the Authors commented in the text, the results cannot exclude the functional deficit observed in Hfe2^{fl/fl};Acta-Cre is due to muscle weakness. To prove this is not the case, Hfe2^{fl/fl};Acta-Cre should be treated with recombinant HJV.

Response: To address this issue, we assessed muscle strength in *Hfe2*^{ΔActa-cre} and *Hfe2*^{fl/fl} mice. We did not observe any difference (falling events and reaching counts) between our 2 groups. The difference observed in behavioral testing is therefore unlikely to be the result of muscle weakness. This novel piece of data is presented in our novel Figure, Extended Data Fig. 6.

We have added technical details for these experiments in the Material and Methods part. Also, we have added the following text to the result section:

“Similarly, *Hfe2*^{ΔActa-cre} animals also displayed functional deficits, these were unlikely to be the result of muscle weakness as the hanging wire strength test did not show any difference between our groups (**Extended Data Fig. 6**).”

3. Since Hfe2 liver KO mice develop severe iron overload, it is unclear why the authors have performed critical experiments in this mouse model instead of using the Hfe2 sk muscle KO that does not display iron overload.

Response: Our original intent was to study the role of liver secreted Hfe2 on blood vessel integrity in the CNS. As noted by the reviewer, Hfe2 secreted by the liver regulates iron loading, to rule out the possibility that the effect that we observed was caused by iron loading, we later used muscle deletion of Hfe2. Muscle deletion of Hfe2 shows leakage as well indicating that Hfe2 is causing leakage independent of the effect on iron loading. The experiment proposed by this reviewer, in which we rescue the phenotype of Hfe2 liver KO with Hfe2 injection, further cements this result (see above).

4. Soluble Hvj sequesters BMPs and decreases BMP-SMAD signaling (Theurl et al., 2011): is the BMP pathway involved in BBB permeability? Is sHvj impairing BBB function by downregulating BMP-SMAD signaling?

Response: To address this possibility, we have looked at BMP canonical signalling in endothelial cells. We looked at psmad1/5/8 activation and we could not observe any difference between RGMa, Hfe2, and controls. We have added this new piece of data in our new Extended Data Fig. 15.

Please note that RGMa and Hfe2 can both work as chelators for BMPs. Since RGMa and Hfe2 both remove BMPs, it is unlikely that this would result in opposite activities on endothelial cells.

The methods part has been modified to integrate p-smad Western Blotting and activation assay. Also, we have added the following text in the results.

“Hfe2 is a Bone Morphogenic Protein (BMP) co-receptor, and RGMa has been shown regulate the blood brain barrier integrity via a BMP-receptor/Yap pathway. To investigate the

mechanisms whereby Hfe2 regulates blood vessel integrity, we tested whether Hfe2 had any effect on the canonical BMP down-stream effector smad1/5/8, however, we could not observe any significant difference between Hfe2 treated cells and controls (**Extended Data Fig. 15**). “

5. The Authors hypothesize that soluble Hvj, by binding to RGMa, prevents the interaction of RGMa with neogenin. Since soluble Hvj binds BMPs, are these ligands necessary to mediate the RGMa- soluble Hvj interaction?

Response: Our protein assay was done in the absence of BMPs. As described in our material and Methods part we tested the interaction between Hfe2 and Neogenin using purified Neogenin and Hfe2.

A paper from Zhang et al., shows that RGMa regulates YAP via the BMP receptor, hence we tested the possibility that Hfe2 regulates YAP activation. In western Blots, we could not observe any change in the levels of p-YAP. Similarly, in a luciferase reporter assay for YAP activation, we could not observe any activation of YAP by Hfe2. We have added this novel piece of data in our new Extended Data Fig. 13.

The methods part has been modified to integrate YAP Western Blotting and activation assay. We also reference the Zhang et al. 2029 article in our manuscript. We have added the following text in the result part;

“Moreover, we assessed YAP activation in a reporter assay and in Western Blots. In these experiments, the addition of Hfe2 to the medium of bEdn3 cells did not lead to any change in YAP expression and activation”.

6. In the EAE model, does the soluble Hvj treatment also affect the inflammatory response? Is soluble Hvj reduced in MS patients?

Response: We have assessed many inflammation markers on immune cell treated with PBS or Hfe2. We could not observe any difference in Hfe2 treated cells when compared to controls. This piece of data is presented in Extended Data Fig. 19.

We are collaborating with Dr. Feng, an MS specialist at the University of Chongqing. He measured Hfe2 (RGMc) in Human controls (HC) and MS patients and his quantification shows a significant reduction of Hfe2 in MS patients.

Dr. Feng would like to use this piece of information in a manuscript that he is working on, while he allowed us to present this data, he does not want it to be part of the present manuscript.

Figure R2: Hfe2/RGMc expression in the plasma of healthy and MS patients. Data from Dr. J. Feng.

7. In skeletal muscle dystrophy, Hfv is reduced (Zhang et al., 2019). Is this accompanied by increased BBB leakage?

Response: This is an interesting point thank you for mentioning it. The paper from Zhang et al., shows that Hfv (Hfe2/RGMc) is reduced. These experiments are done using a model for Duchene muscular dystrophy. Duchene is indeed associated with blood brain barrier leakage. However, genes involved in Duchene disease are also expressed in the brain, and for this reason we cannot link BBB leakages in Duchene to a down-regulation of Hfe2. We have added a section in our discussion to address the role of the muscles in blood vessel maintenance/leakage. We also address the point raised by the reviewer.

“We also show that Hfe2 secreted by muscles play a role in BBB homeostasis. A recent article reveals that Hfe2 is down-regulated in models for Duchene muscular dystrophy (mdx mouse)²². Duchenne muscular dystrophy has been linked to blood barrier breakdown in human patients and mdx mice²². However, dystrophin, the gene involved in Duchenne disease is also expressed in the CNS, where its down-regulation may cause BBB alterations^{22,32}. Therefore, establishing a direct link between BBB changes in Duchenne disease and muscular dystrophy is not feasible. Exercise has many beneficial effects on brain health and helps to restore and maintain cognitive function. It was recently shown that physical exercise regulates cerebral angiogenesis via the release of lactate, which binds to the lactate receptor HCAR1³³. The HCAR1 receptor is highly enriched in

pial fibroblast-like cells, which line the vessels supplying blood to the brain, and in pericyte-like cells, along intracerebral micro-vessels³³. The fact that exercise is sensed by the brain (vasculature) suggests that muscle-induced peripheral factors enable direct crosstalk between muscle and brain function. Our data showing that muscle-Hfe2 promotes BBB protection, may indeed highlight a pivotal element in such crosstalk.”

8. Soluble H₂ is not secreted since it is a GPI-anchored protein but cleaved by furin. Furin is upregulated in iron deficiency-hypoxia. Do the Authors observe if iron deficiency and/or hypoxia are also characterized by BBB leakage, likely due to increased levels of circulating soluble H₂?

Response: This is an interesting point. We have not looked at iron deficiency-hypoxia ourselves. However, a literature search suggests that mild hypoxia induces BBB leakage. Interestingly, hypoxia inducible factor (HIF-1 α) is a known regulator of the blood brain barrier. Because we were interested in identifying factors that may explain the protective effect of Hfe2 on the blood vessels, we decided to evaluate whether Hfe2 and RGMa have any effect on HIF-1 α . We observed that endothelial cell treatment with RGMa increased HIF-1 α expression and that this effect was blocked by the presence of Hfe2.

We are presenting this novel piece of data in our novel Extended Data Fig. 12.

We have modified the Material and Method parts to include experimental details. We have added the following text to the result part:

“Furthermore, we assessed the expression of a few factors that are known to trigger blood vessel dysfunction. Interestingly, we observed that HIF-1 α , a factor that is regulated by hypoxia and triggers BBB opening, is upregulated in the presence of RGMa, and that addition of Hfe2 to the medium blocks this upregulation (**Extended Data Fig. 12**).”

9. The Discussion should be implemented: the results demonstrate that is not the liver but soluble H₂ (released by the liver and the skeletal muscle) that regulates BBB integrity.

Response: We have considerably improved the discussion, which is now over 3 pages long.

REVIEWERS' COMMENTS

Reviewer #1 (Remarks to the Author):

The authors addressed most of my previous concerns. there is only minor point: The authors should explicitly mention that the Tie2Cre driver affects both endothelial cells and hematopoietic lineage cells in the main text, and specifically refer to the additional experiments (extended fig 19 and figure 6e) that partly address the potential influence of Hfe2 on immune cells. The author's additional work to address this concern is appreciated but should be specifically pointed out to the reader in the main text

Reviewer #2 (Remarks to the Author):

The study by Wang, Vigouroux et al. show that liver- and/or muscle-born Hfe2 competes with RGMa in order to maintain BBB maintenance. Within the review they addressed all major concerns from Review1 in a proper way. First of all they added primary human BMECs to support the findings made in mice. Further they checked the influence of iron levels in their KO-model and could erase possible doubts regarding the impact of the KO.

For all minor concerns the authors easily adapted the manuscript and added a new chapter to the short discussion which was necessary. From my side there are no further complaints and the paper is ready to be published.

Reviewer #3 (Remarks to the Author):

Authors did a good job responding to my critique.

I'm supportive of the current manuscript.

Reviewer #4 (Remarks to the Author):

The Authors have addressed my concerns. In addition, the authors have provided new data that further strengthen the role of soluble Hvj in regulating the integrity of Central Nervous System blood vessels. The discussion has also been improved.

We were very pleased to see that our manuscript was accepted with a very minor revision. Reviewer #1 recommended to add some text, which was added as suggested. Other reviewers recommended publications as it is.

A detailed response can be found below.

REVIEWERS' COMMENTS

Reviewer #1 (Remarks to the Author):

The authors addressed most of my previous concerns. There is only one minor point: The authors should explicitly mention that the Tie2Cre driver affects both endothelial cells and hematopoietic lineage cells in the main text, and specifically refer to the additional experiments (extended fig 19 and figure 6e) that partly address the potential influence of Hfe2 on immune cells. The author's additional work to address this concern is appreciated but should be specifically pointed out to the reader in the main text.

Response: We thank the reviewer and have added the following text to our manuscript.

“Because Tie2 and Neogenin are both expressed by immune cells, we tested whether treatment of immune cells with Hfe2 had any effect on the expression of immune marker. Treatment with Hfe2 had no effect on *i*) the adhesion properties of naïve T and B cells, *ii*) naïve antigen presenting cells, *iii*) naïve immune cell populations, *iv*) on activated immune cells, and *v*) antigen specific immune cells which also indicates that the Hfe2 effect is independent of immune cell priming (**Extended Data Fig. 19a-e**). Together these data suggest that prevention of EAE symptoms in *Neo^{ΔTie2-creERT2}* mice results from a restoration of the BBB and not from an alteration of immune cell activation.”

Note that Fig. 6e is already mentioned in the manuscript.

Reviewer #2 (Remarks to the Author):

The study by Wang, Vigouroux et al. shows that liver- and/or muscle-born Hfe2 competes with RGMa in

order to maintain BBB maintenance. Within the review they addressed all major concerns from Review1 in a proper way. First of all they added primary human BMECs to support the findings made in mice. Further they checked the influence of iron levels in their KO-model and could erase possible doubts regarding the impact of the KO.

For all minor concerns the authors easily adapted the manuscript and added a new chapter to the short discussion which was necessary. From my side there are no further complaints and the paper is ready to be published.

Response: We thank the reviewer for his contribution to the review process.

Reviewer #3 (Remarks to the Author):

Authors did a good job responding to my critique.

I'm supportive of the current manuscript.

Response: We thank the reviewer for his contribution to the review process.

Reviewer #4 (Remarks to the Author):

The Authors have addressed my concerns. In addition, the authors have provided new data that further strengthen the role of soluble H₂O₂ in regulating the integrity of Central Nervous System blood vessels. The discussion has also been improved.

Response: We thank the reviewer for his contribution to the review process.